# WHEN MLLMs MEET COMPRESSION DISTORTION: A CODING PARADIGM TAILORED TO MLLMs

**Jinming Liu**[1,2,*], **Zhaoyang Jia**[3,*], **Jiahao Li**[3], **Bin Li**[3], **Xin Jin**[2], **Wenjun Zeng**[2], **Yan Lu**[3]
[1]Shanghai Jiao Tong University     [2]Eastern Institute of Technology, Ningbo, China
[3]Microsoft Research Asia
jmliu206@sjtu.edu.cn    jinxin@eitech.edu.cn
{t-zjia, li.jiahao, libin, yanlu}@microsoft.com

## ABSTRACT

The increasing deployment of powerful Multimodal Large Language Models (MLLMs), typically hosted on cloud platforms, urgently requires effective compression techniques to efficiently transmit signal inputs (e.g., images, videos) from edge devices with minimal bandwidth usage. However, conventional image codecs are optimized for fidelity to serve the Human Visual System (HVS) and ill-suited for MLLMs, in which diverse downstream tasks are jointly considered. In this paper, we first systematically analyze the impact of compression artifacts on several mainstream MLLMs. We find that: *Compression distortion unevenly impacts different-level image features, leading to varying effects on MLLMs' downstream tasks depending on their feature-level reliance.* Motivated by this discovery, we propose an image *Co*dec *TA*ilored to *M*LLMs (CoTAM) designed to adaptively protect multi-level features and suit different demands of downstream tasks. The encoder leverages CLIP's shallow-layer attention to generate an importance map for bit allocation, preserving critical semantic regions. Concurrently, the decoder integrates a lightweight adapter with a multi-level loss function to ensure the faithful reconstruction both of low-level details and high-level semantic context for robust synthesis of cross-level features. Extensive experiments validate that our method achieves up to 35.99% bitrate saving while maintaining the same performance on the MLLM tasks, outperforming previous SOTA neural codecs. The code is released at https://github.com/jmliu206/CoTAM.

## 1 INTRODUCTION

The proliferation of MLLMs, such as GPT-4o Hurst et al. (2024), Gemini Team et al. (2023), and LLaVA Liu et al. (2023a), has marked a paradigm shift in artificial intelligence, revolutionizing human-machine interaction, content understanding Li et al. (2023a), and automation Yin et al. (2024). These models possess an insatiable appetite for high-quality visual data Zhu et al. (2025) to fuel their powerful capabilities. As MLLM applications become ubiquitous—from real-time visual question answering on mobile devices to complex scene analysis in cloud-based services—the demand for transmitting and storing image and video data is growing at an explosive rate. This surge creates a critical bottleneck: *the conflict between the need for high-fidelity visual input and the constraints of limited communication bandwidth and storage resources.* Consequently, developing highly efficient compression techniques tailored for this new era is not just beneficial but imperative.

However, existing compression techniques are ill-suited for the versatile, open-world nature of MLLMs. Conventional codecs are engineered for the HVS Wallace (1991); He et al. (2022); Li et al. (2024c), while Image Coding for Machine (ICM) methods target specific, narrow computer vision tasks Feng et al. (2022); Chamain et al. (2021). This misalignment leads to inconsistent performance across the diverse capabilities of MLLMs. As illustrated in Fig. 1(a)(b), both methods exhibit erratic performance, excelling in some tasks while failing in others He et al. (2022); Kao et al. (2025). Fundamentally, these approaches do not address the crucial question of how MLLMs holistically perceive and are affected by compression artifacts.

---

*Jinming Liu and Zhaoyang Jia are visiting students at Microsoft Research Asia.

To address this gap, our work begins with a systematic investigation into this question. Our analysis reveals a crucial insight: *Compression distortion unevenly impacts different-level image features, leading to varying effects on MLLMs' downstream tasks depending on their feature-level reliance.* Specifically, as shown in Fig. 1(c)(d), our analysis reveals that: tasks that rely on either low-level structural features (e.g., large-font OCR) or global high-level semantic features (e.g., overall scene understanding) both demonstrate relatively robust to compressed distortion. In contrast, tasks requiring a synthesis of cross-level features (e.g., counting objects) are highly susceptible, as compression artifacts disrupt the crucial integration of low-level information into a coherent high-level semantic. Motivated by this finding, we introduce an image Codec TAilored to MLLMs (CoTAM). At the encoder, our codec leverages priors from the shallow layers of a pre-trained CLIP model Radford et al. (2021) to guide rate allocation. At the decoder, a lightweight adapter with the reconstruction prior and a multi-level objective function ensures that both low-level fidelity and high-level perception are faithfully restored. This mechanism resolves the conflicting demands of different task types, ensuring the reconstructed output is faithful to the MLLMs' needs. The main contributions of this work are summarized as follows:

- We first provide a systematic analysis of MLLM performance under compression, revealing how MLLMs are affected by compression distortion.

- We propose CoTAM whose encoder uses lightweight CLIP-based semantic priors for rate allocation while the decoder uses a multi-level loss and adapter with reconstruction priors to preserve multi-level information.

- Our approach achieve significant bitrate savings while delivering consistently high performance across a wide spectrum of MLLM tasks, and also shows compatibility with high-resolution and video-based MLLM scenarios.

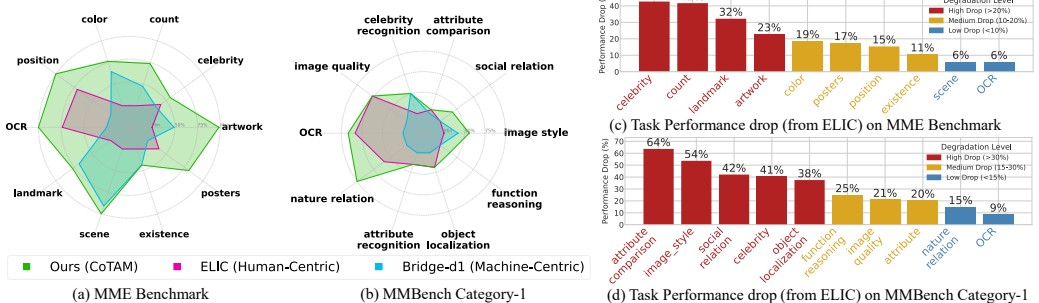

(a) MME Benchmark     (b) MMBench Category-1     (d) Task Performance drop (from ELIC) on MMBench Category-1

Figure 1: (a)(b) Performance comparison of compression methods on MME Fu et al. (2023) and MMBench Liu et al. (2024b) under similar bitrates. For MMBench, we report the 10 most affected tasks (largest score drops). Human-centric codec ELIC He et al. (2022) excels on low-level structural tasks (e.g., Large-font OCR) and the ICM method Bridge-d1 Kao et al. (2025) excels on high-level tasks (e.g., landmark identification), while our method consistently outperforms both. (c)(d) Compression distortion (from ELIC) affects tasks differently: Tasks relying on either low-level structural features or coarse high-level semantics (e.g., OCR and scene understanding) tend to be relatively robust, whereas those depending on cross-level features (e.g., counting) suffer more, reflecting a synthesis that fails when corrupted low-level information can no longer be coherently structured by the high-level context. Seeing more benchmarks' sub-tasks and images in Appendix.

## 2 IMPACT ANALYSIS OF IMAGE DISTORTION ON MLLMS

### 2.1 PRELIMINARIES: THE MLLM PIPELINE

The architecture of mainstream MLLMs Li et al. (2024a); Zhu et al. (2025) comprises three key parts: a vision encoder Radford et al. (2021); Zhai et al. (2023), a projector, and a LLM Bai et al. (2023); Touvron et al. (2023). The vision encoder, often a Vision Transformer (ViT) Han et al. (2022), serves as the model's "eye", responsible for transforming an input image into a sequence of vision tokens. These vision tokens are then passed through the projector (e.g., some MLP layers), a lightweight network that maps them into the LLM's feature space. Finally, the LLM backbone (e.g.,

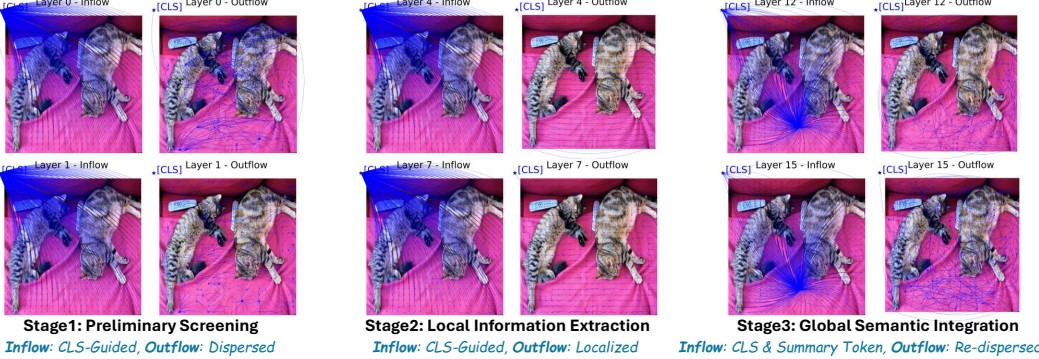

Figure 2: How compression affects VQA tasks: while the MLLM's robust low-level structural and coarse-grained high-level semantic abilities enable it to identify the "strawberry" and "AI", it fails on tasks demanding fine-grained cross-level information, such as providing an accurate count.

Figure 3: Information flow across different layers of the vision encoder. According to the flows, we can divide the information processing to three stages as Table 1. In stage 3, a token with high attention no longer represents its own local visual content, but instead transforms into a high-level 'summary token' Liu et al. (2025); Li et al. (2023c) responsible for integrating global information.

LLama Touvron et al. (2023), Qwen Team (2024)) processes these projected vision tokens alongside a text prompt to perform cross-modal reasoning and generate the final output.

This work investigates the effects of the compression distortion on the Vision Encoder. Because it serves as the sole gateway for visual information into the MLLM, the quality of its output tokens directly dictates the upper bound on the entire model's downstream performance. To this end, in this work, we isolate our analysis from the LLM backbone, whose behavior is conditioned on specific textual prompts, in order to derive general conclusions about the visual processing pipeline itself.

## 2.2 EXPLORING THE IMPACT OF IMAGE COMPRESSION DISTORTION TO MLLMs

To design a codec tailored to MLLMs, we must first understand what visual information MLLMs require and how this information acquisition process is affected by compression artifacts.

### 2.2.1 HOW DOES VISUAL INFORMATION FLOW IN MLLMs?

Prior work shows that weak high-level semantic capability in MLLMs induces hallucinations Fu et al. (2023), whereas supplying richer, clearer image details substantially improves performance Liu et al. (2024a). Together, these findings imply that strong MLLMs must exploit both low-level cues and high-level semantics. A scan of mainstream benchmarks (MME Fu et al. (2023), MMBench Liu et al. (2024b), SEED-Bench Li et al. (2023a)) confirms this breadth: tasks span object recognition and counting, spatial reasoning, OCR, compositional inference, and the interpretation of abstract concepts like emotion and intent. The diversity of these tasks also indicates that *MLLMs rely on visual information across multiple levels of granularity—from low-level pixel details to high-level semantic abstractions.* For example, in Fig. 2, answering "What is the word?" requires low-level structural OCR capabilities, determining "What is this fruit?" demands high-level global semantic reasoning, while the response to "How many strawberries are in this picture?" needs both structural information and global semantics. This raises a pivotal question: *how does the vision encoder transform raw pixels into a feature representation that balances both low-level details and high-level semantics?* To investigate this, we analyze the information flow within the vision encoder (CLIP Radford et al. (2021)), inspired by the inflow/outflow methodology of Tong et al. (2025). Specifically, for a self-attention map $A \in \mathbb{R}^{N \times N}$ in a given layer, where $A_{ji}$ denotes the attention

Table 1: The Three-Stage Pattern of Visual Information Processing in the Vision Encoder.

| Stage | Information Flow (Fig. 3) | [CLS] Attention (Fig. 4(a)(b)) | PCA-Visualized Features (Fig. 4(c)) |
|---|---|---|---|
| **Stage 1:** Preliminary Screening (Shallow Layers) | **Inflow:** Receives global guidance from [CLS]. **Outflow:** Scattered, performs a broad initial screening. | Broad, with higher intensity on key areas. | Resemble raw textures and edges; no significant aggregation. |
| **Stage 2:** Local Information Extraction (Middle Layers) | **Inflow:** Remains anchored to the [CLS] token. **Outflow:** Concentrates on neighboring patches. | Converges on certain edges and local regions. | Extracts low-level features with clear structures. |
| **Stage 3:** Global Semantic Integration (Deep Layers) | **Inflow:** Diversifies, from [CLS] and summary tokens. **Outflow:** Disperses again to integrate refined features. | Converges on a few summary tokens. | Extracts cross-level to abstract, high-level semantics; structural details are discarded. |

Figure 4: Three stages' CLS attention maps and PCA features in Table 1 (layer 0, 5, 15, 22 in vision encoder) for (a) the raw image and (b) the compressed image. (c) The visualization of cosine similarity between raw tokens and distorted tokens. Simiarity is lowest at Early Stage 3, indicating a significant impact on cross-level features.

from source token $i$ to target token $j$, we define two metrics **Information Inflow & Outflow** to trace the primary information pathways. $\text{Inflow}(k) = \arg\max_j A_{kj}$, and $\text{Outflow}(k) = \arg\max_i A_{ik}$.

The visualization of this information flow (Fig. 3) reveals a distinct three-stage feature processing pattern, which is detailed in Table 1 and corroborated by the PCA Maćkiewicz & Ratajczak (1993) visualization and [CLS] attention maps in Fig. 4(a). The process begins with **Stage 1: Preliminary Screening**, where shallow layers perform a broad, initial scan of the image, with attention scattered to capture raw textures and edges. This is followed by **Stage 2: Local Information Extraction**, where middle layers consolidate these findings; the Outflow becomes shorter, with attention converging on neighboring patches to analyze local features with clear structures. Finally, the deep layers execute **Stage 3: Global Semantic Integration**. In this phase, the model integrates refined local features into a holistic, semantic representation, with attention converging on a few key "summary tokens." Liu et al. (2025); Li et al. (2023c)

To quantitatively validate our three-stage finding, we measure two layer-wise attention distance metrics Dosovitskiy et al. (2020) on 1,000 images from the CC3M dataset Changpinyo et al. (2021): the Average Attention Distance ($D_{\text{avg}}$) and the Average Max Attention Distance ($D_{\text{top1}}$).

$$D_{\text{avg}} = \frac{1}{N} \sum_{i=1}^{N} \sum_{j=1}^{N} A_{ij} \cdot d(p_i, p_j), \qquad D_{\text{top1}} = \frac{1}{N} \sum_{i=1}^{N} d(p_i, p_{\arg\max_j A_{ij}})$$

Here, $A$ is the $NN$ attention map from a self-attention layer, where $A_{ij}$ is the attention weight from token $j$ to token $i$. The term $p_i$ denotes the 2D spatial position of the $i$-th token in the input image. Consequently, $d(p_i, p_j)$ represents the Euclidean distance between the positions of tokens $i$ and $j$. As plotted in Fig. 5(a)(b), both metrics exhibit a clear U-shaped trend. The average distance is high during **Stage 1**, decreases for **Stage 2**, and increases again during **Stage 3**. This quantitative trend strongly corroborates our findings.

### 2.2.2 How Does Compression Distortion Affect MLLMs?

Having established the three-stage information flow model, we analyze its vulnerability to compression distortion. By measuring the cosine similarity of feature tokens between original and compressed images at each layer, a clear pattern emerges, as shown in Fig. 5(c). While the low-level features in Stage 1 and 2 prove relatively robust to compression, linearly and slowly decrease in similarity layer by layer, we observe a sharp drop in similarity in the early phase of Stage 3, which marks a critical failure in the formation of cross-level features. These features are uniquely vulnerable because their creation requires a delicate synthesis of high-fidelity low-level details from Stage

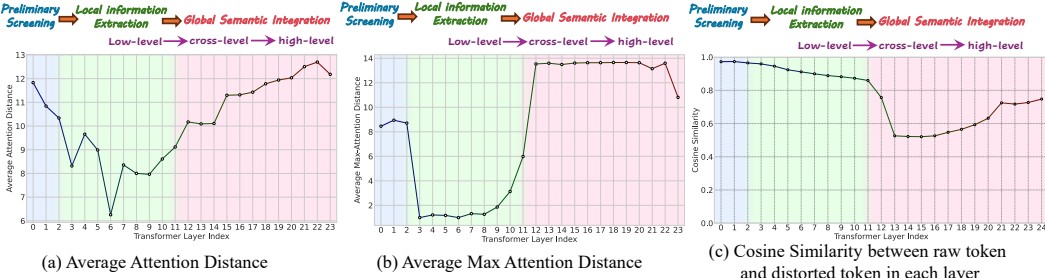

Figure 5: (a)(b) Attention distance and (c) the impact of distortion on internal tokens in the vision encoder. Low-level (Stage 2) and coarse high-level features (the later phase of Stage 3) are relatively robust to compression artifacts, while cross-level features (the early phase of Stage 3) are significantly affected because they require both high-fidelity low-level details and emerging high-level semantic context. Blue, green, and red indicate stages 1, 2, and 3, respectively.

2 and emerging high-level semantic context from Stage 3. Consequently, even the subtle corruption of the low-level details by compression leads to a disproportionately large failure in this synthesis process. In contrast, the similarity recovers in the later part of Stage 3, demonstrating that coarse, high-level semantics are more resilient. This finding is further corroborated by the attention maps, PCA features and cosine similarity in Fig 4(b)(c). While these visualizations show little change in PCA features and high cosine similarity in stages 1 and 2 between original and compressed images, the token similarity in the early phase of stage 3 is significantly decreased. The later part of stage 3 is aimed at generating coarse high-level semantic information. Therefore, the impact of distorted details is diminished, resulting in a higher overall cosine similarity. As shown in Fig. 2, compression distortion only minimally affects questions of high-level semantics (e.g., "What is the fruit?") or low-level structure (e.g., "What is the word?"). Its impact is much greater, however, on tasks like counting or texture analysis, which demand both local details and global context.

Task-level validation confirms this hypothesis. As shown in Fig. 1(c)(d), tasks requiring the synthesis of both detailed and semantic information (e.g., "count") degrade severely under compression. Conversely, tasks reliant on either robust low-level structures (OCR) or coarse high-level semantics (positional reasoning) remain resilient. This leads to a key insight: **the critical failure point of compression is not a uniform loss of different feature types, but a disproportionate collapse of the cross-level representations that bridge low-level and high-level information.**

The takeaways of the above analysis are the following:

> **Takeaways:**
>
> *1. MLLMs require visual information at different levels to perform diverse tasks.*
>
> *2. The vision encoder in MLLMs operates in three stages: shallow layers handle initial filtering, middle layers extract low-level features via local analysis, and deep layers perform global semantic integration, sequentially assembling these features into cross-level and then high-level semantic representations.*
>
> *3. Compression-induced information loss increases linearly in early layers, indicating that low-level features suffer only modest degradation. However, this compromises cross-level features, which rely on integrating low-level information with high-level context to preserve fine-grained semantics. In contrast, coarse high-level features are moderately affected, as they depend more on abstract representations.*

These expose a fundamental paradox in current ICM approaches Kao et al. (2025); Li et al. (2024b); Chamain et al. (2021). They only try to preserve the high-level information but ignore the low-level information, which is important for MLLMs to generate cross-level features. Our work is thus built upon a new cornerstone: **An effective codec must simultaneously preserve proper both low-level fidelity and high-level semantic information.**

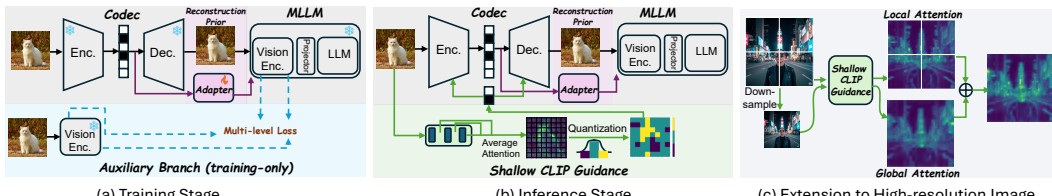

(a) Training Stage.      (b) Inference Stage.      (c) Extension to High-resolution Image.

Figure 6: The framework of our method.

# 3    CoTaM: Codec Tailored to MLLMs

Our analysis reveals a core principle for a codec tailored to MLLMs: it must preserve multi-level visual information. Based on this principle, we introduce CoTaM, a codec designed with a dual-strategy approach, as depicted in Fig. 6(b). First, drawing upon the insight from Takeaway 2—that the initial layers of a vision encoder perform preliminary information filtering—our encoder uses shallow CLIP attention to guide bitrate allocation, prioritizing important regions for MLLMs. Second, inspired by Takeaways 1 and 3, our decoder uses the decompressed image as a reconstruction prior to retain robust low-level details and avoid domain shift. A latent feature adapter then injects semantic enhancements, and the entire model is optimized with a multi-level loss that supervises fidelity at multi-level features. Furthermore, for high-resolution inputs, CoTaM incorporates a Hierarchical Guidance mechanism to fuse multi-scale semantic information, making it compatible with the patch-based processing Liu et al. (2024a) common in MLLMs for both images and videos.

## 3.1    Base Codec

Our base codec enables variable bitrates by adapting the multi-quantizer methodology from Jia et al. (2025); Cui et al. (2021). We equip its internal layers with multiple sets of learned quantization vectors for each bitrate to adaptively allocate bits for each spatial location. This allows the semantic importance map to select a specific vector for each region, thereby assigning more bits to critical areas and fewer to the rest areas. Further architectural details are provided in the Appendix.

## 3.2    Shallow CLIP-Guided Encoder

Our Shallow CLIP-guided encoder is born from the **Takeaways 2** of our prior analysis: the shallow layers of an MLLM's vision encoder perform a preliminary screening to identify regions of potential importance. To leverage this early-stage intelligence, we average the [CLS] attention scores from the first three layers of a frozen CLIP model Radford et al. (2021)—chosen for their high attention distance (Fig. 5)—to create a small downsampled spatial map (e.g., 8x8), which quantifies the semantic richness of each region.

This continuous map is subsequently converted into a discrete, three-level mask via a statistics-based quantization method $\mu \pm k\sigma$. The three integer levels in this mask directly correspond to rate allocation instructions: decrease bitrate, maintain base bitrate, or increase bitrate. Crucially, due to the small size of this map and its quantization into only three values, the bitrate overhead for this map is negligible (128 bits for 336x336 input). This final mask then directly modulates the quantization parameters of our learned compression backbone on a patch-wise basis, ensuring that semantically critical regions for MLLMs are allocated more bitrate and with higher fidelity.

## 3.3    Multi-Level Fidelity Decoder

Our analysis revealed a critical flaw in existing ICM methods: in their pursuit of high-level semantic fidelity, they often degrade the low-level structured information, and also in turn lead to a significant loss of cross-level features. To resolve this problem, our decoder is designed to preserve fidelity across the entire feature hierarchy. It achieves this through two key components:

First, our design leverages the decoded image as a reconstruction prior. This approach serves two critical functions. On the one hand, as shown in Fig. 1 and takeaway 3, since standard compression is already effective at preserving robust low-level structures, using the decoded image ensures this foundational information is retained. On the other hand, it mitigates a potential domain shift, as

MLLM vision encoders are pre-trained on natural RGB images; providing the decoded image as a prior grounds the input in the expected domain. Upon this prior, a lightweight **Latent Feature Adapter**, composed of a single transformer block, operates directly on the decoded latent code from the bitstream. It generates a semantic enhancement feature that is fused (via element-wise addition) with the patch embeddings extracted from the decoded image. This strategy injects high-level guidance directly into the feature domain without disrupting the crucial low-level information.

Second, as illustrated in Fig. 6(a), the entire framework is trained end-to-end using a multi-level fidelity loss, $\mathcal{L}_{\text{total}}$, to supervise the fidelity at both ends of the feature spectrum. This loss is a weighted sum of two components:

$$\mathcal{L}_{\text{total}} = \lambda_{\text{low}}\mathcal{L}_{\text{low}} + \lambda_{\text{high}}\mathcal{L}_{\text{high}} \tag{1}$$

The first component, the **low-level fidelity loss ($\mathcal{L}_{\text{low}}$)**, is designed to preserve fine-grained details often damaged by existing methods. Guided by our finding in **Takeaway 3**, it imposes critical constraints on the shallow layers by minimizing the Mean Squared Error (MSE) between the patch embedding features of the original and decoded images. Simultaneously, the **high-level perceptual loss ($\mathcal{L}_{\text{high}}$)** ensures global semantic coherence by minimizing the MSE between the final-layer token representations of the original and our processed output.

### 3.4 Extension to High-Resolution and Video Inputs

Handling high-resolution images is a critical capability for MLLMs, making it imperative for codecs to support them efficiently. This presents a core dilemma. On one hand, guidance from a single, fixed-size downsampled image is too coarse; as shown in Fig. 6(c), the background attention is relatively coarse, failing to focus on important information. A direct strategy to adapt to this, inspired by mainstream MLLM processing pipelines, is to employ a patch-based method where local guidance is applied to each patch independently. The fundamental limitation of this approach, however, is its lack of global perception; it cannot determine which local information is crucial for building coherent semantics across different patches. For instance, in Fig. 6(c), it lacks sufficient attention on the person's head. Therefore, to resolve this conflict between local detail preservation and global semantic integrity, we propose our **Hierarchical Guidance** to fuse (via addition) both global and local maps, creating a comprehensive guidance signal that is both locally precise and globally aware. On the other hand, we resize the decoded high-resolution features to get a global feature before they are processed by the adapter. This is done to match the expected input of the high-resolution MLLM, which is composed of multiple high-resolution patches and a downsampled global patch.

Our method is also compatible with video MLLMs. Current mainstream approaches typically process videos by sampling a sequence of individual frames, a strategy analogous to the patch-based processing of high-resolution images. Consequently, our semantic guidance mechanism can be applied on a frame-by-frame basis to guide the compression of videos.

## 4 Experiment

### 4.1 Experimental Settings

**Codec Setting.** Our framework is built upon two learned image compression models, ELIC He et al. (2022) and DCAE Lu et al. (2025) to demonstrate the versatility of our approach in being integrated with different codecs. For model training, we utilized a dataset comprising one million images randomly sampled from the CC3M dataset Changpinyo et al. (2021). The training protocol spans a total of five epochs, with the first epoch dedicated to an initialization phase using only the low-level fidelity loss ($\mathcal{L}_{low}$). This pre-training step ensures a stable optimization trajectory by allowing the network to first grasp the reconstruction of basic structural features. For hyperparameters $k$, $\lambda_{low}$ and $\lambda_{high}$, we empirically set them to 0.75, 0.1, and 1, respectively.

**MLLM Setting.** For MLLM evaluation, our primary experiments were conducted on LLaVA-1.5 Liu et al. (2024a)(both 7B and 13B variants with a CLIP encoder Radford et al. (2021)) to assess performance and scalability. To further substantiate the generalization capabilities of our method, we also performed tests on LLaVA-Onevision-7B Li et al. (2024a) (with a SigLIP encoder Zhai et al. (2023)) and InternVL2-8B Chen et al. (2024) (with an InternViT encoder Gao et al. (2024)).

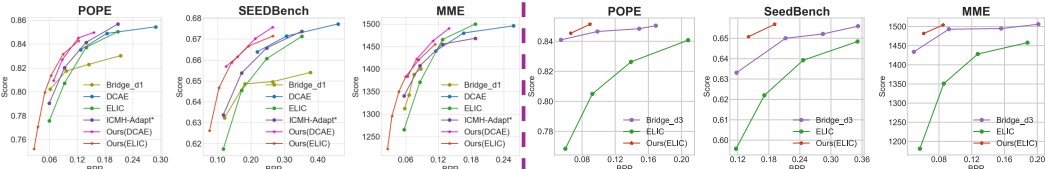

Figure 7: Performance comparison on LLaVA-1.5-7B.

Figure 8: Left three: Performance comparison on LLaVA-1.5-13B. Right three: Performance comparison with methods that fine-tuned the codec encoder.

**Testing Benchmark** Our evaluation protocol is twofold, assessing both MLLM tasks performance and image reconstruction quality. For image benchmark, we evaluated on MME Fu et al. (2023), TextVQA Singh et al. (2019), POPE Li et al. (2023b), SeedBench Li et al. (2023a), VQAv2 Goyal et al. (2017), MMMU Yue et al. (2024), and MMBench Liu et al. (2024b). For video benchmark: we used Video-MME Fu et al. (2025). For reconstruction metric, we report PSNR.

**Compared Methods** To position our work within the current landscape, we compared the codec against a comprehensive set of baselines. For human-centric image compression methods, we selected ELIC He et al. (2022), and DACE Lu et al. (2025). For coding for machine methods, we compared against Bridge-d1 (fixing encoder), Bridge-d3 (finetuning encoder) Kao et al. (2025) and ICMH-adapt Li et al. (2024b). Since ICMH-adapt Li et al. (2024b) only supports the ResNet architecture, we reimplemented this method and trained it with our multi-level loss.

## 4.2 PERFORMANCE COMPARISON

### 4.2.1 LOW-RESOLUTION IMAGE BENCHMARK

Our primary validation, presented in Fig. 7, is conducted on the LLaVA-v1.5-7B model with a 336x336 input resolution. Using ELIC as the base codec, our method consistently outperforms previous approaches across six diverse benchmarks. As shown in Table 2, under the same performance level, it achieves a 35.99% bitrate saving. To demonstrate its generalizability, we integrated our method with another SOTA codec, DCAE Lu et al. (2025), and achieved similar performance gains. The scalability of our approach is further validated in Fig. 8, where we also show improvements on the larger LLaVA-1.5-13B model, proving its effectiveness across different model scales.

**Finetuning Codec.** While our main approach freezes the codec to sidestep the performance–reconstruction trade-off, we also test a fine-tuning variant by adding a rate loss Minnen et al. (2018) to the objective (Eq. 1). The results, presented in Fig. 8, show that even in this comparison with another fine-tuning method Kao et al. (2025), our approach demonstrates superior performance. Furthermore, both methods significantly outperform the original, non-fine-tuned base codec.

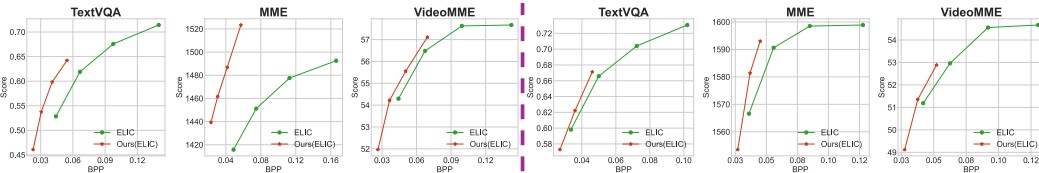

Figure 9: Performance comparison on High-resolution and Video MLLM. Left three: LLaVA-Onevision-7B. Right three: InternVL2-8B.

### 4.2.2 HIGH-RESOLUTION IMAGE AND VIDEO BENCHMARK

Addressing the significant overhead of high-resolution data, we extend our method to this domain. To the best of our knowledge, our work is the first to pioneer a coding framework for high-resolution image and video MLLMs. We validate this on two mainstream models, LLaVA-OneVision-7B and InternVL2-8B, with results presented in Fig. 9. For high-resolution images, our approach consistently outperforms the base codec. Because the current mainstream Video LLM usually extracts video frames into fixed frame images (such as 16, 32 frames), our method can also be directly applied to video MLLM. The codec also achieves superior performance on Video-MME.

### 4.3 ABLATION STUDY

**Framework.** To assess each component's contribution, we perform an ablation study. As shown in Fig. 10(a)(b)(c), the removal of the Adapter module induces a catastrophic degradation in performance across all three benchmarks. This consistent and vast performance underscores the Adapter's role as an essential bridge between the compressed features and the downstream MLLM; its function in aligning feature spaces is both indispensable and universally critical.

Conversely, ablating the image reconstruction module (blue curve) also impacts performance, but with varying severity across benchmarks, reflecting different dependencies on visual fidelity. For TextVQA (Fig. 10(a)) and SeedBench (Fig. 10(c)), "Ours (w/o Rec.)" drops sharply relative to the full model, highlighting the value of reconstruction-induced prior knowledge. In contrast, the impact on MME (Fig. 10(b)) is much milder.

Lastly, removing the clip guidance module (brown curve) consistently reduces performance across benchmarks, indicating it as an effective general optimization.

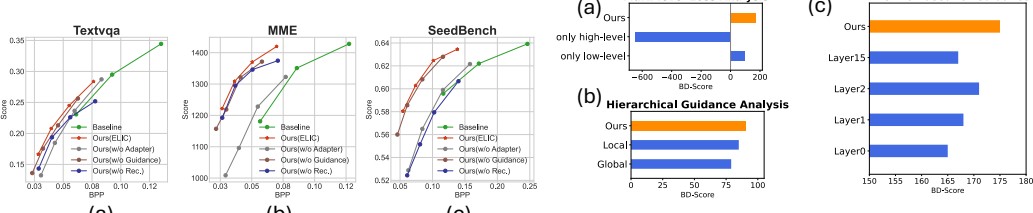

Figure 10: Ablation study on framework.

Figure 11: Ablation study on modules. We test BD-score using ELIC as the anchor on MME.

**Training Loss.** We validate the necessity of our multi-level loss design. As shown in Fig. 11(a), relying solely on the high-level loss fails to capture essential low-level details, while using only the low-level loss produces detailed yet semantically inconsistent results. Optimal performance is achieved by integrating both.

**Hierarchical Guidance.** For high-resolution images, our proposed Hierarchical Guidance improves the importance map by fusing local and global attention. The results in Fig. 11(b) demonstrate that it yields a clear performance improvement over a purely global guidance strategy.

**Attention Maps.** Our use of averaged attention maps from CLIP's first three layers is validated in Fig. 11(c), achieving optimal performance as shallow layers are better for holistic screening. In contrast, deeper layers emphasize global aggregation and thus degrade performance, consistent with our three-stage information flow model.

### 4.4 COMPLEXITY ANYLSIS

We analyze the computational complexity of our method in Table 2. Since our approach only utilizes the first three shallow layers of the CLIP encoder, the increase in encoding time is marginal compared to the base codec. Furthermore, as our framework does not require fine-tuning the codec and the CLIP guidance only reallocates bit rates, the overall PSNR in Fig. 12 shows only a minor degradation compared to the base codec.

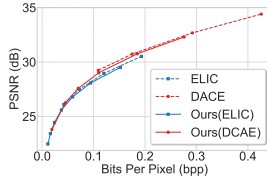

Figure 12: PSNR comparison on Kodak dataset.

| Method | Encoding (s) | Decoding (s) | Total (s) | BD-Rate↓ |
|---|---|---|---|---|
| ELIC | 0.173 | 0.096 | 0.269 | 0.00 |
| **Ours (ELIC)** | 0.178 (+2.9%) | 0.101 (+5.2%) | 0.279 (+3.7%) | **-35.99%** |
| DCAE | 0.077 | 0.085 | 0.162 | 0.00 |
| **Ours (DCAE)** | 0.080 (+3.9%) | 0.091 (+7.1%) | 0.171 (+5.6%) | **-31.05%** |

Table 2: Comparison of times on Kodak dataset (resized as 336x336), and average BD-rate on six MLLM benchmarks, which represents the bitrate saved to achieve the same score.

## 5 CONCLUSION

We conduct a comprehensive analysis of how compression artifacts affect MLLMs, revealing that fine-grained semantic features in cross-level features are highly vulnerable to subtle low-level distortions. Based on this insight, we propose a codec tailored to MLLMs, featuring CLIP-guided bit allocation and a multi-level fidelity preserved decoder. Our method consistently achieves significant bitrate savings while preserving MLLM performance across diverse tasks. This work underscores the importance of compression strategies aligned with the feature hierarchy of MLLMs.

## 6 ACKNOWLEDGMENTS

This work was supported by Grants of NSFC 62302246, ZJNSFC LQ23F010008, Ningbo 2023Z237 & 2024Z284 & 2024Z289 & 2023CX050011 & 2025Z038 & 2025Z059, and supported by High Performance Computing Center at Eastern Institute of Technology and Ningbo Institute of Digital Twin.

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

# A   APPENDIX

## A.1   RELATED WORKS

### A.1.1   MULTIMODAL LARGE LANGUAGE MODELS (MLLMs)

Multimodal Large Language Models (MLLMs), such as LLaVA Liu et al. (2023a), Gemini Team et al. (2023), and GPT-4o Hurst et al. (2024), have demonstrated remarkable capabilities by augmenting Large Language Models (LLMs) with visual perception. These models typically use a vision encoder (e.g., Clip Radford et al. (2021), SigLip Zhai et al. (2023)) to process images and an LLM backbone (e.g., LLama Touvron et al. (2023), Qwen Bai et al. (2023)) to perform cross-modal reasoning. However, the prevailing cloud-edge deployment of MLLMs—hosting powerful models on servers while capturing data at the edge—presents a significant communication bottleneck. This challenge motivates our work to develop a compression solution optimized not for human viewing, but for the unique perceptual needs of MLLMs.

### A.1.2   IMAGE COMPRESSION

The fundamental goal of image compression is to minimize the bits required to represent an image—thereby reducing storage and transmission costs—while maintaining sufficient fidelity for its intended application. Conventional image compression, encompassing both traditional standards like JPEG and VVC Wallace (1991); Bross et al. (2021), and modern learned methods Liu et al. (2023b); Lu et al. (2025), is fundamentally optimized for the Human Visual System (HVS) Li et al. (2025), often by discarding information that is imperceptible to humans but potentially vital for machine analysis.

To bridge the gap created by this human-centric paradigm, the field of Image Coding for Machine (ICM) emerged. However, the predominant ICM approach Feng et al. (2022); Chen et al. (2023); Li et al. (2024b) involves tailoring codecs for narrow, specific tasks like object detection or segmentation. However, this task-specificity is fundamentally at odds with the general-purpose nature of MLLMs. Thus, a critical research gap remains for a compression solution that preserves the full visual features required by these models.

## A.2   BENCHMARK EXAMPLES

To illustrate the diversity of tasks that Multimodal Large Language Models are expected to perform, we provide representative examples from two key benchmarks used in our evaluation. Fig. 13 and 14 showcase selected question-answer pairs from the MME Fu et al. (2023) and SEED-Bench Li et al. (2023a) benchmarks, respectively. These tasks range from object recognition and counting to

Optical Character Recognition (OCR). Notably, the OCR examples involve large-font text where understanding the overall structure and positional relationships is crucial for correct interpretation. Furthermore, the examples highlight the varied question formats MLLMs must address, encompassing both binary (Yes/No) judgments and multiple-choice selections. Collectively, these examples underscore the necessity for a compression codec to preserve a wide spectrum of visual information—from fine-grained details and high-level semantics to the essential structural and positional cues required by these diverse tasks. This challenge is a core motivation for our work.

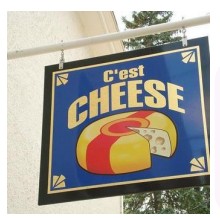

**OCR**

Is the word in the logo \"c'est cheese\"?
Answer the question using a single word or phrase.

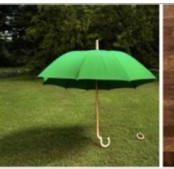

**Commonsense Reasoning**

I want to carry one thing with me on a rainy day. Is the thing in the image an appropriate choice?
Answer the question using a single word or phrase.

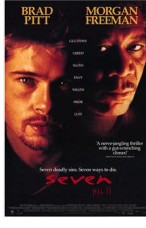

**Posters**

"Is this movie titled a perfect world (1993)?
Answer the question using a single word or phrase."

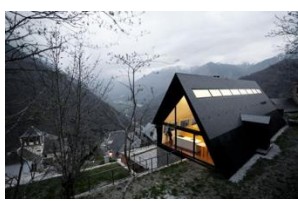

**Scene**

Does this image describe a place of chalet?
Answer the question using a single word or phrase.

Figure 13: The QA pair examples in MME Benchmark Fu et al. (2023).

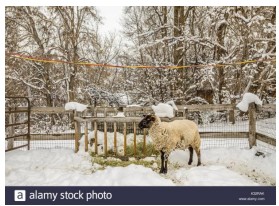

**Scene Understanding**

What is the dominant color in the image?
A. White
B. Brown
C. Black
D. Grey
Answer with the option's letter from the given choices directly.

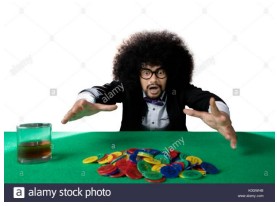

**Visual Reasoning**

What is the most likely reasoning behind the man's shocked expression?
A. He got a bad grade on a test
B. He suddenly remembered a forgotten appointment
C. He's witnessing a fight
D. He didn't expect the amount of poker chips on the table
Answer with the option's letter from the given choices directly.

Figure 14: The QA pair examples in SeedBench Benchmark Li et al. (2023a).

## A.3 CODEC TRAINING STRATEGY

Following Jia et al. (2025); Cui et al. (2021), our codec is trained to operate at multiple bitrates within a single, unified model architecture, as shown in Fig. 15. The core of this variable-rate capability lies in the integration of learnable vectors at multiple intermediate layers of the encoder. These vectors perform a scaling of the feature maps to dynamically control the information flow and, consequently, the final rate-distortion trade-off.

Let the feature map at the output of the $l$-th encoder layer be denoted as $f_l$. For a discrete set of $N$ target bitrates $\mathcal{R} = r_1, r_2, \ldots, r_N$, we introduce $N$ corresponding sets of learnable vectors. For

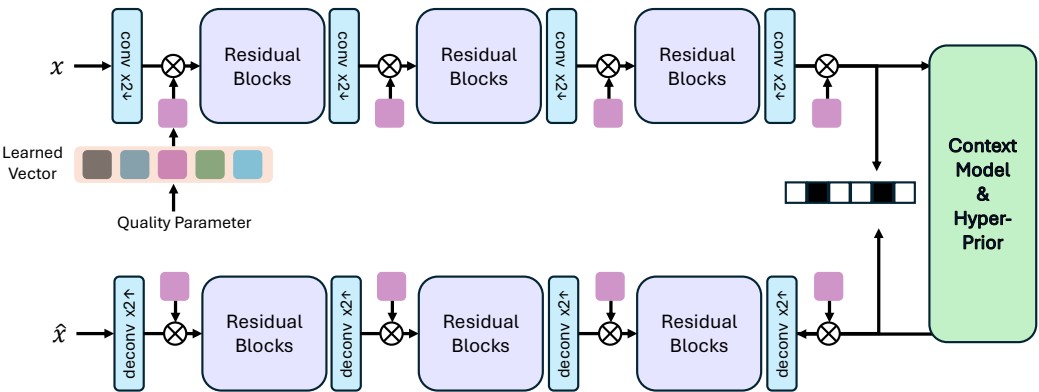

Figure 15: The variable-bitrates compression frameworks.

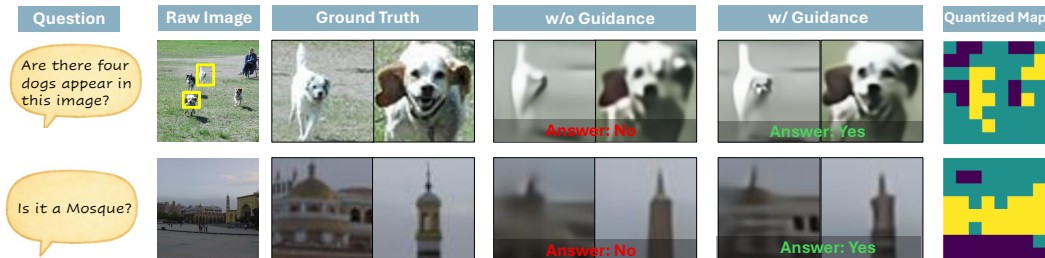

Figure 16: The visualization results under similar total bitrates.

a given target rate $r \in \mathcal{R}$, a specific vector $\boldsymbol{g}_{l,r}$ is applied to the feature map $\boldsymbol{f}_l$ at each modulated layer $l$. This operation is formulated as:

$$\boldsymbol{f}'_l = \boldsymbol{f}_l \odot \boldsymbol{g}_{l,r} \tag{2}$$

where $\odot$ represents element-wise multiplication, and $\boldsymbol{f}'_l$ is the scaled feature map that serves as the input to the subsequent layer $l+1$.

During the training process, a quality index $i$ is randomly sampled in each iteration. This determines both the set of gain vectors $\boldsymbol{g}_{l,i}$ to be used in the forward pass and the corresponding trade-off parameter $\lambda_i$ for the loss function. The entire network, including all $N$ sets of gain vectors, is optimized end-to-end using the rate-distortion loss:

$$\mathcal{L} = \mathcal{D} + \lambda_i \mathcal{R} \tag{3}$$

where $\mathcal{D}$ is the distortion loss and $\mathcal{R}$ is the estimated bit rate. By using a different $\lambda_i$ for each quality level (where a larger $\lambda_i$ encourages a lower bitrate). In our implementation, we empirically define $N = 10$ quality levels, with the corresponding set of tradeoff parameters being $\lambda_i \in \{0.00002, 0.00005, 0.0001, 0.0002, 0.0004, 0.0008, 0.0016, 0.0032, 0.0064, 0.0128\}$.

## A.4 VISUALIZATION OF BIT-RATE ALLOCATION

Fig. 16 visualizes the results of our guidance map's bit-rate reallocation. It clearly shows that more bits are allocated to semantically important regions, leading to higher fidelity for objects like dogs in the example.

## A.5 PRELIMINARY EXPERIMENTS ON FINETUNING STRATEGIES

A straightforward strategy to optimize a codec for a MLLM is to directly finetune either the codec or the MLLM on a downstream instruction-following task. To evaluate the efficacy of these approaches, we conducted a set of preliminary experiments. We employed the LLaVA-Instruct dataset Liu et al. (2023a) for finetuning, using a standard cross-entropy loss in MLLM as the optimization objective.

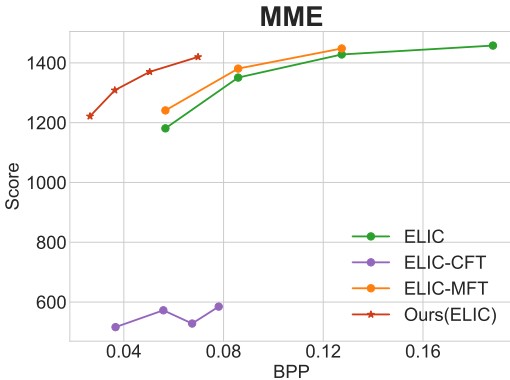

Figure 17: Preliminary experiments on finetuning strategies. ELIC-MFT indicates that the codec parameters are frozen and the MLLM is finetuned. For ELIC-CFT, only the codec parameters are finetuned. Our method only need to finetune the adapter.

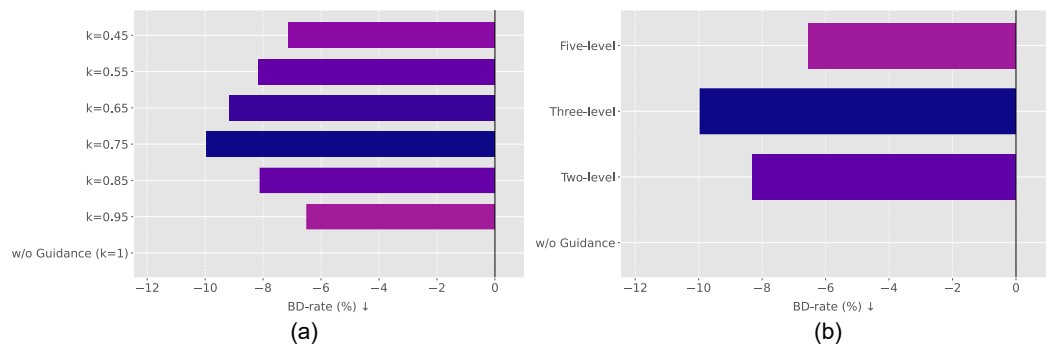

Figure 18: (a) Effect of the statistics-based guidance quantization parameter $k$ in $\mu \pm k\sigma$ on BD-rate. All tested values improve over the no-guidance baseline, with performance remaining stable for $k \in [0.45, 0.85]$. (b) Impact of the number of quantization levels in the Guidance Quantized Map on BD-rate.

As illustrated in Figure 17, our findings reveal the limitations of direct finetuning. When the codec parameters are frozen and the MLLM is finetuned (ELIC-MFT), we observe only marginal performance gains. More strikingly, when we freeze the MLLM and attempt to finetune the codec (ELIC-CFT), the training process collapses, leading to a catastrophic failure where the model loses its fundamental comprehension abilities. In stark contrast, our proposed method, which only requires finetuning a lightweight adapter, yields substantial performance improvements. These results underscore the inadequacy of direct finetuning and motivate our approach.

## A.6 DISCUSSION ON THE QUANTIZED CLIP GUIDANCE MAP

We explored multiple settings of the Quantized CLIP Guidance Map. First, we examined the statistics-based quantization method $\mu \pm k\sigma$ with different values of $k$. As shown in Fig. 18(a), all $k$ values yield improvements, and the performance varies only slightly within the range $k \in [0.45, 0.85]$. Although we adopt $k = 0.75$ in the paper, other values within this suitable range produce similar results. Furthermore, as shown in Fig. 18(b), we investigated different numbers of quantization levels. While using multiple levels generally improves performance, the gain diminishes when the spacing between levels becomes large (e.g., the five-level setting), possibly because the wider span assigns overly low quality to some regions, thereby reducing overall performance.

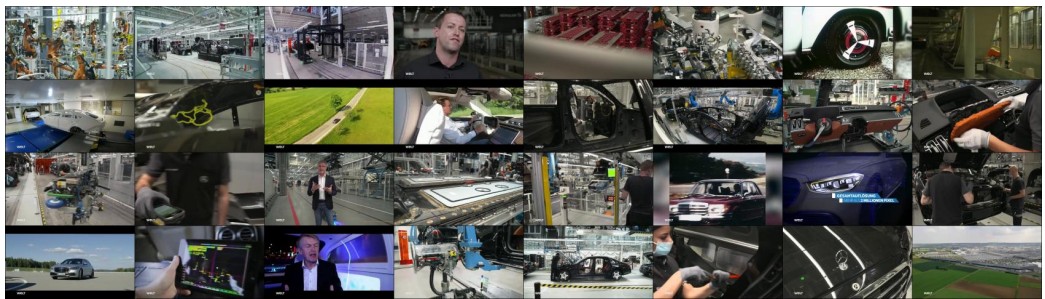

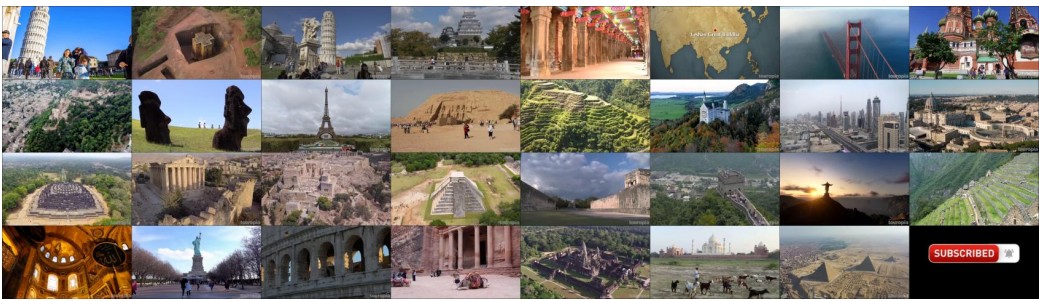

Figure 19: The input examples of video MLLM.

## A.7 DISCUSSION ON VIDEO MLLM

Our method is also directly applicable to video MLLMs. We note that current mainstream video MLLMs, such as Li et al. (2024a); Chen et al. (2024); Zhu et al. (2025); Hurst et al. (2024), typically operate not on dense video streams, but on a sparsely sampled sequence of keyframes (e.g., 16 or 32 frames extracted from the entire video), as shown in Fig. 19. This sparse sampling strategy inherently reduces the temporal redundancy between adjacent processed frames. Therefore, applying our image codec on a frame-by-frame basis is a practical and well-aligned strategy for this specific application. Consequently, our semantic guidance mechanism can be effectively applied to each sampled frame to guide the compression. While developing a more advanced video codec that explicitly models the remaining long-range temporal correlations presents a valuable direction for future work, our current intra-frame approach offers a strong and pragmatic baseline for compressing visual inputs for today's video MLLMs.

## A.8 ATTENTION DISTANCE OF DIFFERENT VISION ENCODERS

To validate that our three-stage information flow model is a general principle rather than an artifact of a specific architecture, we extend our analysis to other prominent vision encoders, namely Intern-ViT Chen et al. (2024) in InternVL2 and SigLIP Zhai et al. (2023) in LLaVA-Onevision Li et al. (2024a). As shown in Fig. 20, the average attention distance per layer for both encoders exhibits a clear U-shaped trend, mirroring the pattern observed with the CLIP encoder in our main analysis. This corroborates our finding that vision encoders broadly follow a three-stage process: an initial broad screening (Stage 1), followed by localized feature extraction (Stage 2), and concluding with global semantic integration (Stage 3). Furthermore, Fig. 21 reveals that the feature similarity under compression shows a sharp drop during the early phase of Stage 3, followed by a recovery in the final layers. This behavior is consistent with our three-stage theory: the initial drop highlights the vulnerability of cross-level features during the synthesis of local details and emerging global context, while the subsequent rebound indicates the formation of a more stable, abstract semantic representation. Notably, SigLIP exhibits two sharp drops in similarity. We hypothesize this is due to SigLIP's architecture, which lacks a dedicated class token. Consequently, its deeper layers may need to retain some local information for the final pooling process, leading to a lower overall feature similarity. Nevertheless, the feature similarity in SigLIP's final layer still rebounds, which remains consistent with the global semantic integration phase of the three-stage process.

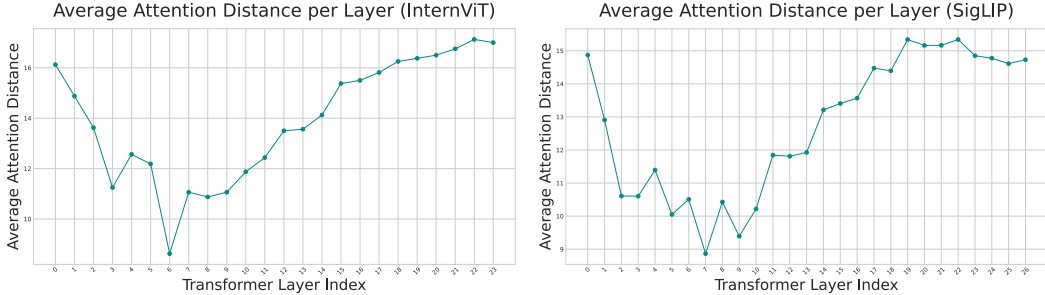

Figure 20: The average attention distance per layer of different vision encoder (InternViT and SigLIP).

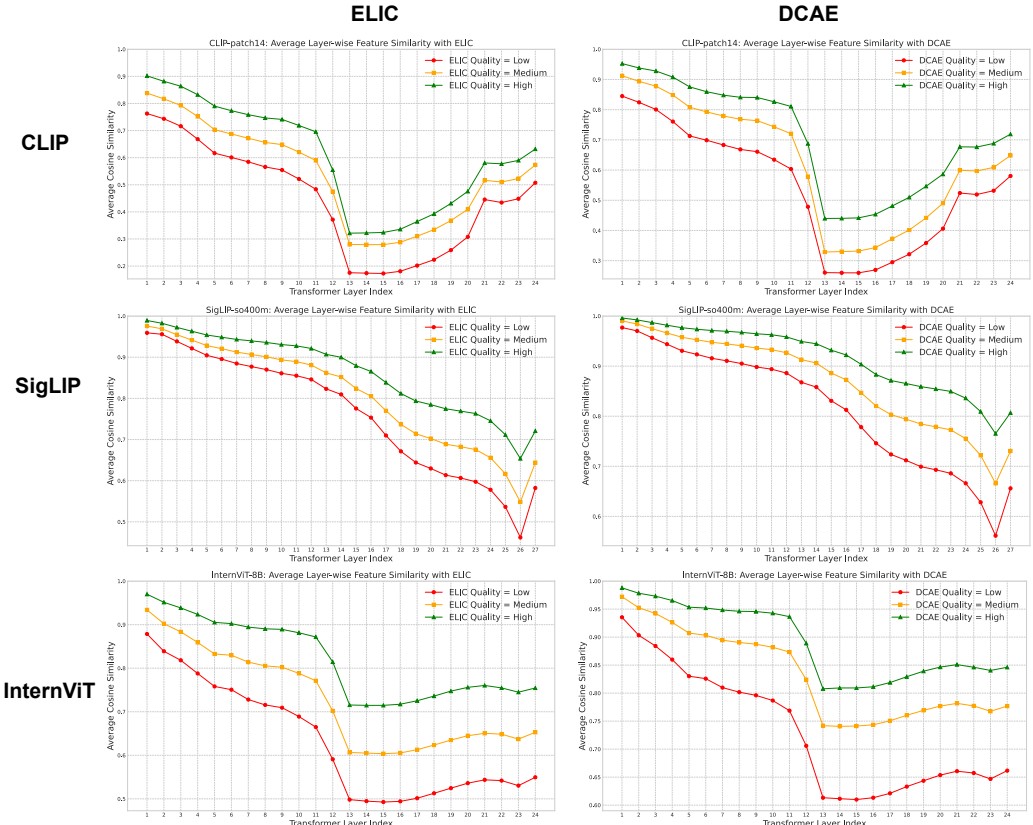

Figure 21: The impact (token similarity) of distortion on internal tokens in the vision encoder.

### A.9 ICM METHOD TASK-WISE PERFORMANCE DROP LEADED BY COMPRESSION DISTORTION

To provide a more granular view of the inconsistent performance of existing codecs, we present a detailed task-wise breakdown of the performance degradation caused by compression. Fig. 22 shows the impact of ELIC, a codec optimized for human perception, on various sub-tasks within the MMBench benchmark. Fig. 23 further illustrates the performance drop on both MME and MM-Bench when using Bridge-d1, an Image Coding for Machine (ICM) method. These figures highlight that both human-centric and machine-centric codecs exhibit erratic performance, excelling in some task categories while failing significantly in others. This inconsistency reinforces the argument that a new paradigm is needed—one that is holistically tailored to the multi-level feature requirements of general-purpose MLLMs.

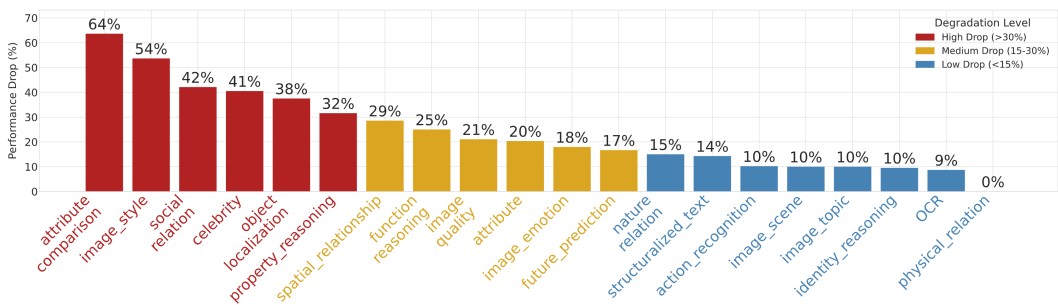

Figure 22: The task-wise impact of compression distortion (from ELIC) on MMBench.

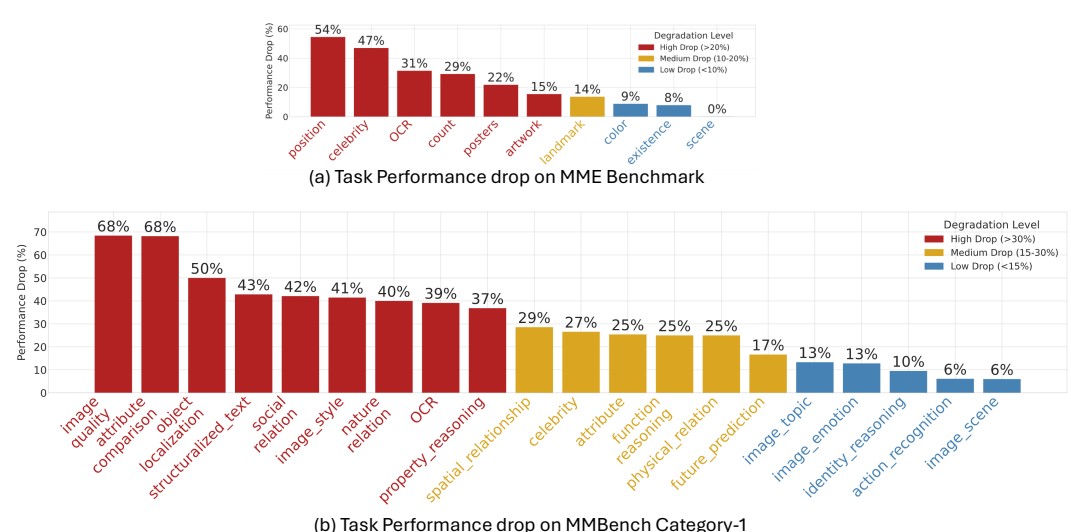

(a) Task Performance drop on MME Benchmark

(b) Task Performance drop on MMBench Category-1

Figure 23: The task-wise impact of compression distortion (from ICM method Bridge-d1) on MME and MMBench.

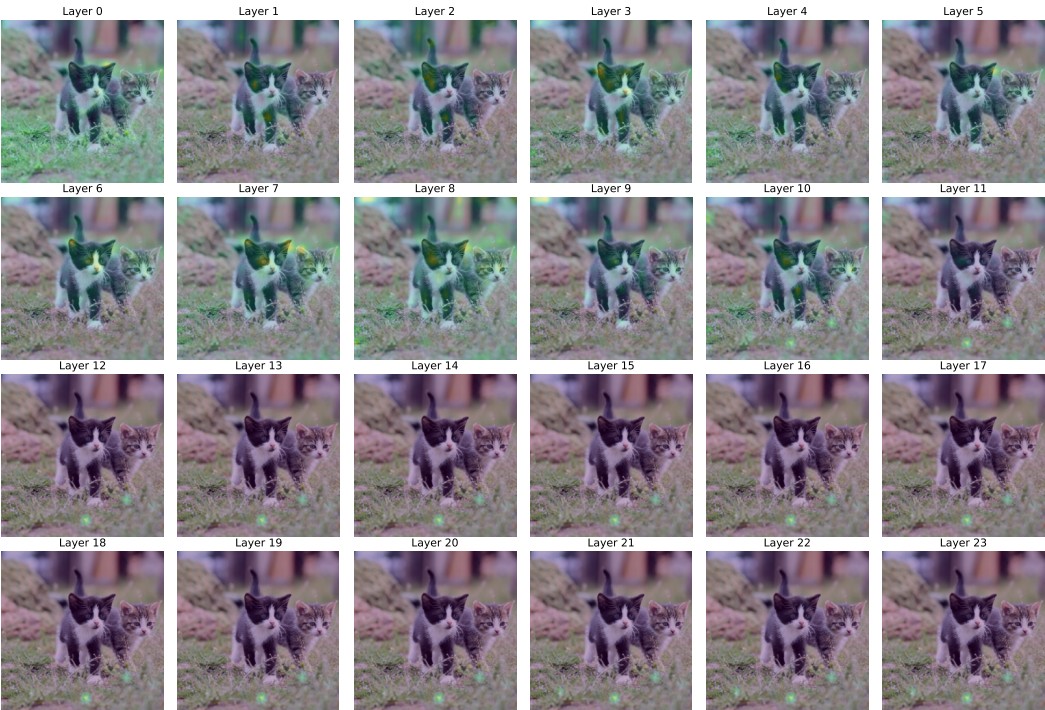

Figure 24: The attention maps of the class token in different layers.

## A.10 ATTENTION MAP AND PCA FEATURES OF DIFFERENT LAYERS

Fig. 24 provides a visual walkthrough of the [CLS] token's attention maps at different layers of the vision encoder, substantiating our three-stage model. In the shallow layers (Stage 1, e.g., layer 0), the attention is broad and scattered, performing a preliminary screening of the entire image. As we move to the middle layers (Stage 2, e.g., layer 7), attention becomes more focused, converging on local regions and edges to extract structured features. Finally, in the deep layers (Stage 3, e.g., layer 22), the attention disperses again as the model integrates globally aggregated information, with focus shifting to a few "summary tokens" that encapsulate high-level semantic concepts.

Complementing this, the Principal Component Analysis (PCA) visualizations in Fig. 25 reveal the evolution of the features themselves. Features in the shallow layers resemble raw textures and edges. In the middle layers, these evolve into clearly structured local features. By the time we reach the deep layers, the structural details are largely discarded in favor of abstract, high-level semantic representations. Together, the attention patterns and the feature visualizations provide strong, complementary evidence for the distinct information processing stages within the vision encoder.

## A.11 INFORMATION FLOW OF DIFFERENT LAYER

To further dissect the information processing dynamics, we analyze the inflow and outflow patterns for tokens across different layers, as illustrated in Fig. 26. This analysis reveals a clear three-stage progression. Initially, in Stage 1, tokens exhibit wide-ranging inflow and outflow without a clear focus, a pattern characteristic of a broad initial screening of the image. Subsequently, Stage 2 is marked by an asymmetric information flow: inflow remains anchored to the global [CLS] token for guidance, while outflow becomes highly localized to neighboring patches, reflecting a focus on structured local feature extraction. Finally, in Stage 3, the dynamics shift again as most tokens receive targeted inflow from a few "summary" tokens, which in turn broadcast their globally-integrated semantic knowledge via global outflow. This dynamic confirms the final phase of semantic synthesis and integration.

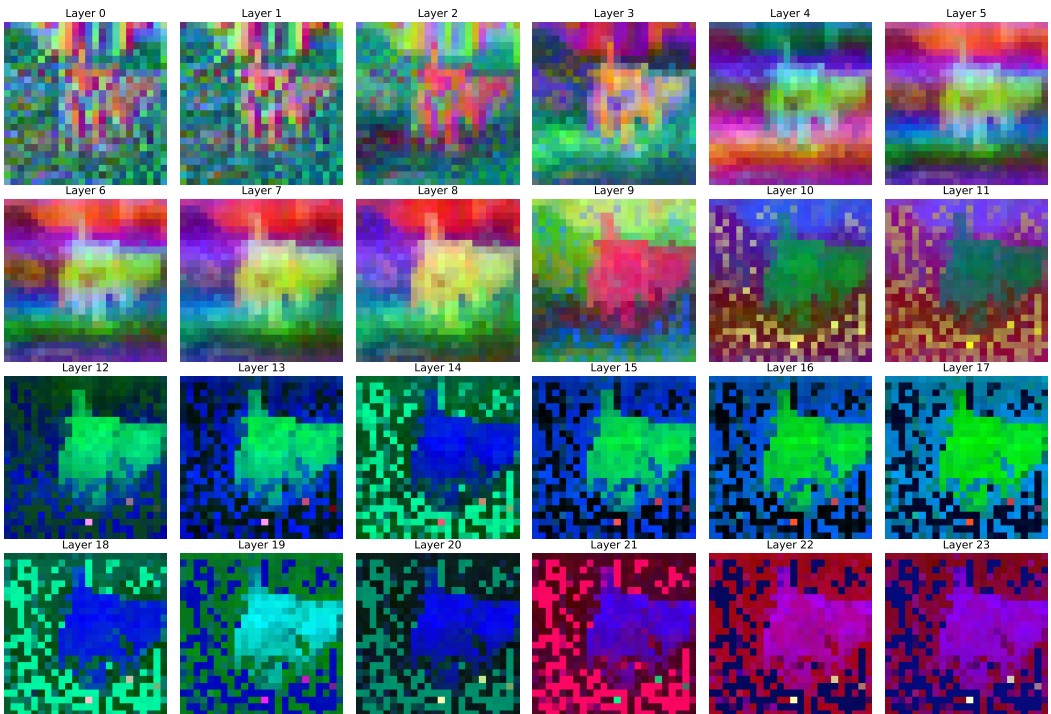

Figure 25: The PCA features in different layers.

## A.12 ANALYSIS ON OTHER DATASETS

In Fig. 27, we present the results obtained from analyzing different datasets. It can be observed that the general curve trends are consistent with those in Fig. 5, indicating that different datasets do not affect our conclusion.

## A.13 LLM USAGE

We acknowledge the use of a large language model (LLM) to assist in the preparation of this manuscript. The LLM's role was strictly limited to improving grammar and refining language. It did not contribute to any of the core research components, such as the initial ideas, experimental design, data analysis, or interpretation of the results.

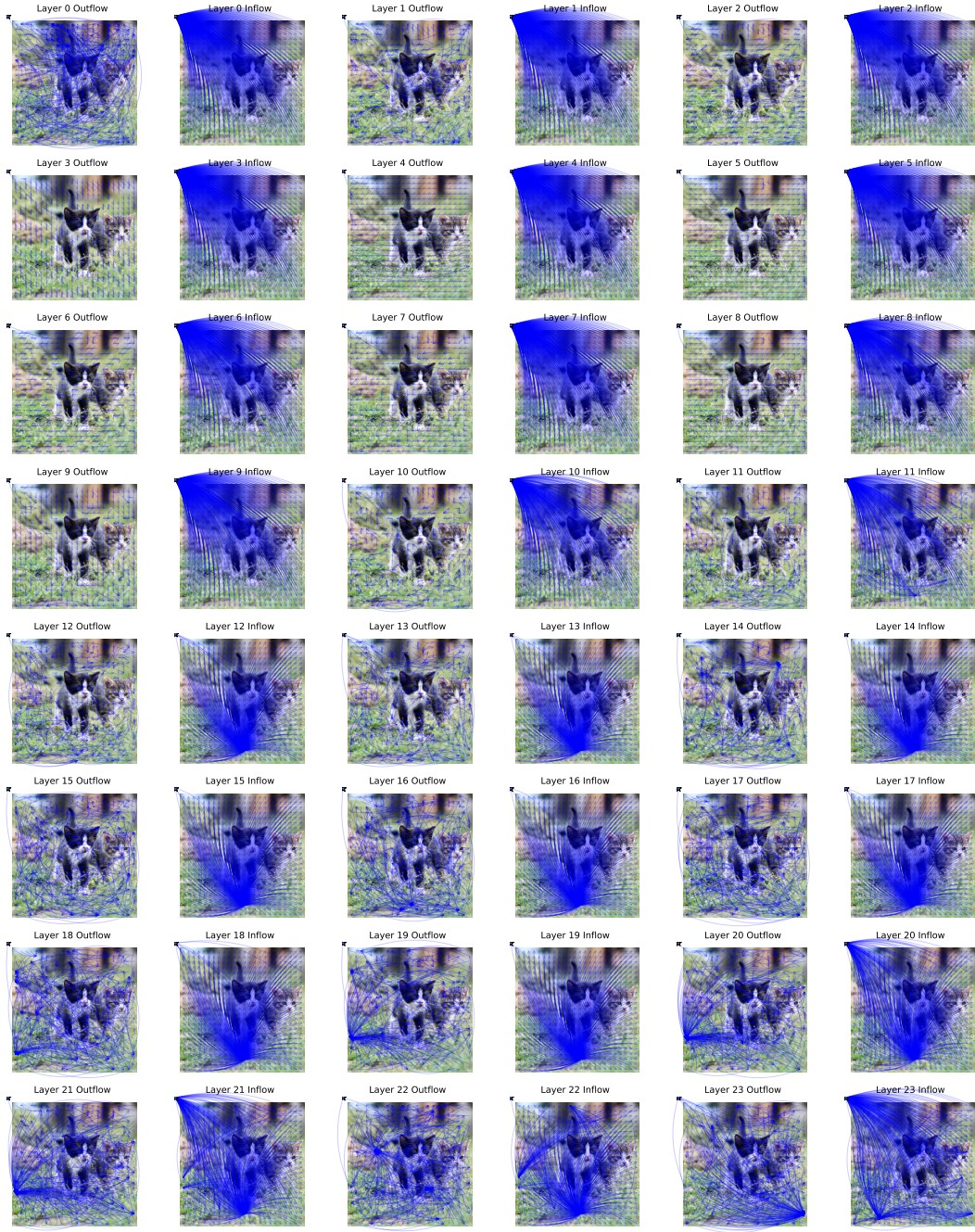

Figure 26: The information flows of different layers.

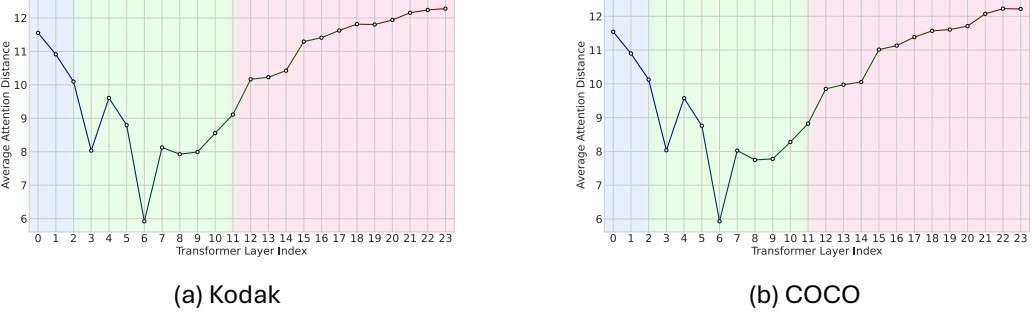

(a) Kodak               (b) COCO

Figure 27: The attention distance on different datasets.

