# OpenReview forum: "When MLLMs Meet Compression Distortion: A Coding Paradigm Tailored to MLLMs"
_ICLR.cc/2026/Conference — ICLR 2026 Poster_

### Official Review · Reviewer_g6n1 · 2025-10-24

**Soundness:** 3
**Presentation:** 2
**Contribution:** 3
**Rating:** 4
**Confidence:** 4

**Summary:**

This paper notes that cloud-hosted Multimodal Large Language Models (MLLMs) urgently need efficient compression for edge-device image/video transmission with low bandwidth. Conventional codecs, optimized for Human Visual System (HVS) fidelity, are ill-suited for MLLMs’ diverse downstream tasks. Through analysis, the authors find compression distortion unevenly affects image features, impacting MLLMs based on their feature reliance. They propose CoTAM, an MLLM-tailored codec: its encoder uses CLIP’s shallow attention for bit-allocation (preserving key semantics), while its decoder employs a lightweight adapter and multi-level loss to reconstruct low-level details and high-level context. Experiments show CoTAM cuts bitrate by up to 35.99% while maintaining MLLM performance, outperforming prior SOTA neural codecs.

**Strengths:**

1. Clear and concise description of the proposed method.

2. It depicts the influence of compression distortion on the performance of MLLM, and analyzes the change between original and compressed image as shown in Figure 5.

3. It can achieve the better results within TextVQA, MME and VideoMME tasks.

4. "a disproportionate collapse of the cross-level representations that bridge low-level and high-level information", a interesting observation and may be the key to further improve the performance of MLLMs.

**Weaknesses:**

1. LLM backbones maybe matter, but this manuscript doesn't compare different LLM backbones.

2. The Equation about $D_{avg}$ and $D_{top1}$ lack the detailed explanation for $A, d and p_i$.

3. The average of $[CLS]$ attention scores lacks the motivation, and the detailed calculation of average score is not provided. Besides the other tokens in CLIP may work well even better.

4. The global attention of high-resolution images in Figure 6(c) is not appropriate since the ratio of downsample is not clear, and how to downsample the image is not to provided.

5. Traditional Codec methods are not compared with the proposed method though the $Adapter$ in Figure 6(b) is not in these traditional methods.

**Questions:**

1. The clarification of $[CLS]$ attention scores may improve the score of the paper.

2. The variety of MLLMs and Codecs may result in different analysis.

---

> ### Author Response · Authors · 2025-11-19
> **Response to Reviewer g6n1 Part I**
>
> We would like to thank the reviewer for taking the time to review our work.
>
> 1. **Different LLM Backbone experiments**
>
>     We would like to clarify that our experiments, in fact, do cover a diverse range of LLM backbones, demonstrating the generalizability of our proposed method.
>
>     The MLLMs selected for our evaluation are built upon distinct LLM architectures. Specifically:
>     * LLaVA-1.5 (used in Fig. 7 and Fig. 8) is based on Vicuna.
>     * LLaVA-Onevision-7B (used in Fig. 9) utilizes Qwen2.
>     * InternVL2-8B (used in Fig. 9) is built on InternLM2.
>
>     Our proposed method, CoTaM, consistently outperforms other compression techniques across all these models, which strongly suggests that its effectiveness is not tied to a specific LLM backbone but rather addresses a more fundamental issue in the visual processing pipeline.
>
>     Meanwhile, as we elaborate in Section 2.1 (lines 106-107), our work's primary focus is on the visual front-end to derive general principles about how compression affects the visual pipeline itself without the effect of languages.
>
> 2. **The Defination of Equation**
>
>     We apologize that our definition of the equation was incomplete; we have corrected this in the revised version. In our original version, the definition of $A$ was on line 156, self-attention. Here, $A$ is the $N×N$ attention map from a self-attention layer, where $A_{ij}$ is the attention weight from token $j$ to token $i$. The term $p_i$ denotes the 2D spatial position of the $i$-th token in the input image. Consequently, $d(p_i, p_j)$ represents the Euclidean distance between the positions of tokens $i$ and $j$. These metrics allow us to quantify the average distance over which information is aggregated in each layer.
>
> 3. **Discusssion with [CLS] Token calculation**
>
> * "Motivation for Averaging": Our analysis in Section 2.2.1 identifies that the vision encoder's shallow layers perform a "preliminary screening." Averaging the attention maps of the first three layers produces a more stable and robust importance map than relying on a single, potentially noisy layer. Our ablation study (Fig. 11(c)) confirms that this approach yields better performance.
>
> * "Calculation": We will clarify in Section 3.2 that we normalize the [CLS] attention maps from the first three layers and then compute their element-wise average.
>
> * "Why [CLS] Token": We use the [CLS] token because it is trained to aggregate global semantic information for image-text alignment in CLIP. In contrast, patch tokens capture more localized dependencies. As our information flow analysis (Fig. 3) shows, the [CLS] token acts as a global guide, making it ideal for our purpose of guiding bit allocation across the entire image.
>
>     We attempt to obtain importance is to calculate the average attention of each patch token receives from all others in the sequence. The results of llava-onevision on MME benchmark are shown below. It can be seen that using the [Patch] token to obtain importance yields lower performance than using the [CLS] token.
>
>     | Method                     | BD-Rate $\downarrow$ |
>     |----------------------------|---------|
>     | w/o Guidance               | 0       |
>     | w/ CLIP [CLS] Guidance     | -6.85 % |
>     | w/ CLIP [Patch] Guidance   | -2.67 % |
>
>
> 4. **Clarify Global Image Donwsampling Method**
>
>     MLLMs like LLaVA-Onevision or InternVL typically process images of fixed size using their vision encoders. Therefore, when handling high-resolution images, they achieve this through "a low-resolution global patch + N high-resolution local patches." These N+1 patches have the same size (e.g., 336x336 or 384x384). Therefore, when extracting global image, we also use the same downsampling method (e.g., bicubic) as LLaVA-Onevision or InternVL to obtain a fixed size low-resolution global patch.

---

> > ### Author Response · Authors · 2025-11-19
> > **Response to Reviewer g6n1 Part II**
> >
> > 5. **Comparison with Traditional Codec**
> >
> >     We further provide BD-rate comparison results (lower is better) between Traditional Codec VTM-18.1 and Ours (ELIC). It can be seen that our method outperforms VTM-18.1.
> >
> >     | Method              | MME     | POPE    |
> >     |--------------|---------|---------|
> >     | ELIC         | 0       | 0       |
> >     | VTM-18.1     | 4.71%   | -5.70%  |
> >     | Ours（ELIC） | -50.22% | -37.39% |
> >
> >
> > 6. **Robustness of the variety of MLLMs and Codecs**
> >
> >     We designed CoTAM as a versatile framework that is not tied to a specific base codec or a single MLLM. Our experiments were structured to prove this robustness.
> >     - Different Codecs: We implemented and evaluated CoTAM on top of two distinct and state-of-the-art learned image codecs: ELIC and DCAE. As demonstrated in our main results in Figure 7 and Table 2, our method brings consistent and significant bitrate savings (e.g., -35.99% for ELIC and -31.05% for DCAE) and performance gains on both backbones. This shows our approach is codec-agnostic.
> >     - Different MLLMs: We evaluated the final performance of our CoTAM-enhanced codecs on three different MLLMs, which use distinct vision encoders: LLaVA-1.5 (7B and 13B variants, using CLIP)，LLaVA-Onevision-7B (using SigLIP)，InternVL2-8B (using InternViT).
> >
> >     Additionally, Figures 20 and 21 in the appendix provide analyses of variety codecs and vision encoders, demonstrating the universality of our theory.

---

> > > ### Comment · Reviewer_g6n1 · 2025-11-24
> > > **Response to authors**
> > >
> > > Thanks for the detailed explanations. I have also read the weaknesses of other reviewers and the authors' responses, and most of the responses have addressed my concerns. Though the results of MME on VTM are not very good and the work targets relatively few downstream tasks, the research is generally solid, so I will raise my score.

---

> > > > ### Author Response · Authors · 2025-11-25
> > > >
> > > > Thank you very much for considering raising your score. We greatly appreciate your constructive feedback throughout the review process.

---

### Official Review · Reviewer_xBDN · 2025-10-27

**Soundness:** 3
**Presentation:** 4
**Contribution:** 4
**Rating:** 8
**Confidence:** 4

**Summary:**

The authors address the problem of image compression for multimodal large language models (MLLMs) in a multi-task setting. They conduct a detailed analysis of how the image compression process impacts features at different levels and draw several interesting and meaningful conclusions. Building on these analyses, they design a new image compression architecture—based on existing work—that improves MLLM task performance.

**Strengths:**

Overall, the idea is interesting, the topic is timely, and the paper is generally well written and well structured.

**Weaknesses:**

I have several concerns regarding the depth and clarity of the analysis section:
- The proposed three-stage theory is intriguing, but the justification appears primarily driven by the observation in Figure 5(b). Although both Figures 5(a) and 5(b) exhibit a U-shaped curve, only Figure 5(b) shows a small plateau at the beginning that supports the notion of three distinct stages. a) Please clarify whether this plateau is the main empirical evidence for defining the three stages. b) The caption of Figure 5 refers to “Average Max Attention Distance,” whereas the text mentions “Distance to the Most-Attended Token.” Are these the same metric, or do they differ? If they differ, please explain the relationship between them. c) Is the precise separation between stages (e.g., between Stage 1 and Stage 2) dependent on the model architecture or the dataset used? Providing a more rigorous or generalizable explanation would strengthen the argument.

- More details should be given regarding the compression methods used. For example, for most compression algorithms—traditional or learning-based—using a fixed quality factor across different images typically results in different bitrates. a) In generating Figure 4, did the authors fix the quality parameter across images, or did they fix the bitrate to ensure comparability? b) The same question applies to subsequent experiments in the experiment section: were all methods evaluated under consistent bitrate conditions? c) Additionally, please clarify whether the observed phenomenon holds across different quality levels or if it appears only under specific compression strengths.

**Questions:**

N/A

---

> ### Author Response · Authors · 2025-11-19
> **Response to Reviewer xBDN**
>
> Thank you for taking the time to review our work and for your recognition of our work. Below are some clarifications regarding your concerns.
>
> 1. **The Justification of Three-stage Theory**
>
>     a) The "Three-stage Theory" relies not only on the "plateau" in Figure 5(b), but also on a comprehensive conclusion combining information flow visualization (Figure 3), attention map visualization (Figure 4), PCA feature evolution (Figure 4), and quantitative measurement (Figure 5). The U-shaped curve (high-low-high) in Figure 5 strongly supports the three stages of "dispersion-focusing-redispersion". We summarize this information in Table 1. More detailed visualization results can be found in Figures 24, 25, and 26 in the appendix.
>
>     b) “Average Max Attention Distance” and “Distance to the Most-Attended Token.” are the same metric. Thank you for your comments, we will modify this.
>
>     c) "Model architecture or the dataset effect for the precise separation between stages:" The experimental results for SigLIP and InternViT in Appendix A.8 of the paper demonstrate that this pattern is universal across different ViT architectures, although the specific layer boundary may vary depending on the model depth. We further added results using different datasets, as shown in Figure 27. The U-shaped curve and the three-stage pattern are still clearly visible, which proves the universality of our theory.
>
> 2. **Bitrate Control**
>
>     a) The single compressed image presented for visualization in Figure 4 is indeed the result of one compression instance and thus has a single, specific bitrate.
>
>     b) When comparing different methods, we use two methods to align bitrates.
>     * In Figure 1, ELIC and bridge-d1 both use the same base codec, therefore their bitrates are also the same. Our method uses clip guidance for bitrate allocation, which deviates slightly from the other two. However, by adjusting the quality factor, we controlled the bitrate difference between our method and the other two methods to 0.01 bpp.
>     * Furthermore, similar to Figures 7 and 8, we plotted the performance at different bitrates for a performance comparison across the wide range. Our method, CoTAM, consistently outperforms the baseline method across the entire bitrate range (from low to high) (as shown in Figure 7).
>
>     Therefore, our comparisons are conducted under rigorously controlled conditions: either at nearly identical bitrates for point-wise comparisons or across the full rate-distortion spectrum for a comprehensive evaluation.
>
>     c) "observed phenomenon holds across different quality levels:"
>     Figure 21 in the appendix shows the trend of token similarity under different compression qualities (Low, Medium, High, corresponding to different bitrates). As can be seen, regardless of the compression quality, a noticeable "sudden drop" appears in early Stage 3 (the stage of cross-level feature formation). This perfectly demonstrates the universality of this phenomenon.

---

> > ### Comment · Reviewer_xBDN · 2025-11-25
> > **Bitrate control across images**
> >
> > Thank you for your response. My initial question about the bitrate control concerns bitrate control across different images rather than across different compression approaches. For a given compression method and a fixed quality level, different images within a dataset naturally produce different bitrates (and different distortions). In your numerical analyses on a dataset, do you use the same quality levels for all images—thus accepting the resulting variations in bitrate—or do you adjust the quality level per image so that the bitrates are approximately aligned?

---

> > > ### Author Response · Authors · 2025-11-25
> > >
> > > Thank you for the clarification. In our experiments, we followed the evaluation protocols established in previous works such as ELIC, DCAE, and Bridge-d1. Specifically, we applied fixed quality parameters across the entire dataset. As a result, the bitrate varies from image to image depending on content complexity. Since our analysis focuses on performance under consistent quality settings rather than a fixed target bitrate, we did not adaptively adjust the quality levels to align the bitrates for each individual image.

---

> > > > ### Comment · Reviewer_xBDN · 2025-11-25
> > > >
> > > > Thank you for the answer. It is important to note that the quality level is an intrinsic parameter of the compression method itself, rather than a parameter that directly dictates the characteristics of each compressed image. As a result, when using a fixed compression approach with a single quality factor, the resulting bitrates—and the corresponding distortions—can vary substantially across images. Since your analysis examines the effect of compression on MLLMs, it is generally preferable to align either the bitrate or the distortion across images. I recognize that this practice is often overlooked in the community, so it is understandable that your paper follows the current convention. Still, it would be helpful to explicitly state the strategy you adopted in your experiments.

---

> > > > > ### Author Response · Authors · 2025-11-25
> > > > >
> > > > > Thank you for the insightful comment and for highlighting this important nuance.
> > > > > We fully agree with your perspective.
> > > > > We will revise the experimental setup section in the manuscript to explicitly state our evaluation strategy.

---

### Official Review · Reviewer_yANt · 2025-10-28

**Soundness:** 3
**Presentation:** 3
**Contribution:** 3
**Rating:** 6
**Confidence:** 4

**Summary:**

This paper designs an image compression codec specifically for Multimodal Large Language Models (MLLMs). The authors argue that existing codecs, optimized for either the Human Visual System (HVS) or narrow machine-vision tasks, are ill-suited for the diverse, general-purpose nature of MLLMs. The paper's contribution is twofold: 1. It first presents an analysis of how compression affects MLLMs, modeling the vision encoder's information flow as a "three-stage" process (Screening, Local Extraction, Global Integration). The key finding is that compression disproportionately degrades "cross-level features"—the synthesis of low-level details and high-level semantics—while robust low-level and coarse high-level features are less affected. 2. Proposed Method (CoTAM): Based on this analysis, the authors propose a codec, CoTAM, which features (a) a "Shallow CLIP Guidance" mechanism at the encoder to allocate more bits to semantically important regions (identified by shallow CLIP-layer attention) and (b) a "Multi-level Fidelity Decoder" that uses a lightweight adapter and a multi-level loss function (supervising both shallow and deep features) to protect the vulnerable cross-level representations. Experiments across a comprehensive suite of MLLM benchmarks (MME, TextVQA, POPE, etc.) show that CoTAM achieves significant bitrate savings (up to 36% BD-rate) over human-centric (ELIC) and machine-centric (Bridge-d1) codecs, all while maintaining MLLM task performance and incurring minimal computational overhead.

**Strengths:**

1. The paper tackles a problem of high practical and academic importance. As MLLMs become central to many applications, developing codecs that are "aware" of their specific perceptual needs is a critical research direction.

2. The core methodology is well-motivated. The idea of first analyzing the internal failure point of MLLMs under compression (the vulnerability of cross-level features) and then designing a codec (multi-level loss, semantic guidance) to specifically protect that failure point is logical and elegant.

3. The experimental validation is a major strength. The authors evaluate on a wide and diverse set of MLLM benchmarks, which is essential for "general-purpose" MLLM compression. The reported BD-rate savings of ~36% (Table 2) are substantial and clearly demonstrate the method's effectiveness over strong, relevant baselines.

**Weaknesses:**

Despite its strengths, the paper has several weaknesses that challenge its claims.

1. The "three-stage information flow" (Sec 2.2.1) is presented as a novel discovery. However, the U-shaped attention distance curve (Fig 5a/b) and the general progression from local, low-level features in shallow layers to global, semantic features in deep layers are well-established properties of Vision Transformers. This part of the analysis feels more like a re-characterization of known ViT properties in the context of compression, rather than a fundamental new insight into MLLMs.

2. The 'Shallow CLIP Guidance' (Sec 3.2) is implementation-specific, creating a dependency that the paper doesn't fully resolve. While the authors evaluate on MLLMs with SigLIP and InternViT encoders , they only demonstrate that CLIP guidance is better than no guidance on a simple ablation. The appendix (A.8) merely shows that these encoders have similar U-shaped attention curves; it does not validate that CLIP's shallow attention is an optimal, or even a valid, importance map for these architecturally different encoders. The study would be significantly strengthened by testing a 'matched-encoder' guidance, such as using a SigLIP-guided map for the SigLIP-based MLLM."

3. The guidance map is an 8x8 grid quantized to only three levels ($\mu \pm k\sigma$). For a 336x336 input, each guidance block covers a 42x42 pixel patch. This seems incredibly coarse. It is difficult to believe this crude map can effectively guide bit allocation for tasks requiring fine-grained detail or small object recognition. The paper needs to provide much stronger justification or analysis for this coarse-grained approach. Maybe add some fine-grained results.

4. I am confused by some experimental results. For example, Figures 7 and 8 show that Bridge d1 is even worse than  ELIC. Can the author explain how the method is tested and which encoder is used?

**Questions:**

Please see the weakness part. Overall, I find the paper satisfying and interesting. The paper would be further improved if the authors addressed the weaknesses above.

---

> ### Author Response · Authors · 2025-11-19
> **Response to Reviewer yANt**
>
> We would like to thank the reviewer for taking the time to review our work.
>
> 1. **Clarify the Contribution of the Three-stage Theory**
>
>     These macroscopic characteristics of ViT (U-shaped attention curve, from local to global) are known conclusions in the field.
>
>     However, our approach differs in that:
>
> * we propose a more refined "three-stage model" than the "local to global" model, and most importantly, we pinpoint the most vulnerable link—the "cross-level feature synthesis" process from stage two to stage three. Simultaneously, the attention maps in the first stage was used to improve task performance.
>
> * We validated the three-stage model from multiple perspectives, including information flow, attention, PCA features, and quantitative analysis.
>
> * We establish a direct link between these internal vision encoder mechanisms and the performance of MLLM on specific tasks under compression distortion.
>
>
> 2. **Why not use matched guidance**
>
>     Because we wanted to build a general approach that wouldn't require redesigning the specific structure of the downstream MLLM, thus offering greater practicality and scalability.
>
>     Since different vision encoders have different structures, using their attention as guidance isn't always feasible. For example, SigLIP lacks a [CLS] token, making it impossible to directly obtain global importance form [CLS]. An alternative method to obtain importance is to calculate the average attention of each [Patch] token receives from all others in the sequence. Tokens with higher average attention are considered more important. Here, we compare the effects of using this method with CLIP-guide, and we can see that CLIP [CLS] actually yields better results because [CLS] better captures global relationships.
>
>     | Method                     | BD-Rate $\downarrow$ |
>     |----------------------------|---------|
>     | w/o Guidance               | 0       |
>     | w/ CLIP [CLS] Guidance     | -6.85 % |
>     | w/ CLIP [Patch] Guidance   | -2.67 % |
>     | w/ SigLIP [Patch] Guidance | -4.89 % |
>
>     We also compared the InternViT-guided map for InternVL, and we can see that the CLIP-guided map and the InternViT-guided map are comparable.
>
>     | Method                      | BD-Rate $\downarrow$ |
>     |-----------------------------|---------|
>     | w/o Guidance                | 0       |
>     | w/ CLIP [CLS] Guidance      | -7.51 % |
>     | w/ InternViT [CLS] Guidance | -8.63 % |
>
> 3. **Clarify for Grid Size**
>
>     MLLM vision encoders are limited by their own granularity (usually one token corresponds to 14 $\times$ 14 or 16 $\times$ 16 pixels), making them challenging for some fine-grained tasks. As shown in the table below, we tested using a larger grid for 336 $\times$ 336 inference on MME Benchmark, but the BD-Rate actually worsened because a larger grid introduces more bpp overhead and the improvement in benchmark score is relatively small. Considering performance and the bit overhead of the maps, we finally chose an 8x8 grid.
>
>     | Grid Size | Bit Overheads | Bpp Overheads | BD-Rate $\downarrow$  |
>     |-----------|---------------|---------------|----------|
>     | 4 $\times$ 4     | 32 bits       | 0.0003        | -45.53 % |
>     | 8 $\times$ 8     | 128 bits      | 0.0010        | -50.18%  |
>     | 12 $\times$ 12   | 288 bits      | 0.0026        | -49.89%  |
>     | 24 $\times$ 24   | 1152 bits     | 0.0102        | -49.51%  |
>
>     You can also view our visualization in Figure 16. The first subplot shows the granularity we can capture, including small objects like people and dogs. For finer-grained tasks, we can refer to the MLLM approach and use a higher resolution.
>
>     We also present the performance data for three fine-grained perception sub-tasks defined in the MME paper, and our method still achieves better results than anchor ELIC on these tasks.
>
>     | Task      | BD-Rate $\downarrow$ |
>     |-----------|---------|
>     | Poster    | -37.15% |
>     | Celebrity | -16.32% |
>     | Landmark  | -40.23% |
>
>
>
> 4. **Explanation of the "Bridge d1" Results**
>
>     We used the author's offical repository and trained Bridge-d1 using ELIC as the base codec and the same vision encoder of our method. As shown in Figure 1, 22 and 23, MLLM requires capabilities at multiple different levels, but Bridge-d1 is only optimized for high-level loss in its paper, it loses some capabilities at other-levels tasks. Therefore, in terms of overall benchmark score, it may even be worse than ELIC. The second sub-figure from the left in Figure 7 also clearly demonstrates this point: on benchmarks such as textVQA_val, which are more focused on low-level tasks, Bridge-d1 performs significantly worse than ELIC. Furthermore, in most benchmarks, Bridge-d1 mainly performs poorly at high bitrates, indicating that optimization for high-level tasks results in the inability to preserve low-level details, leading to a significant upper bound on its performance.

---

> > ### Comment · Reviewer_yANt · 2025-11-26
> >
> > I thank the authors for their detailed response and for conducting the additional experiments I requested. I have reviewed the rebuttal, the revised manuscript, and the responses to other reviewers. I believe my concerns have been addressed.  I will maintain my positive score.

---

### Official Review · Reviewer_iGt4 · 2025-10-31

**Soundness:** 3
**Presentation:** 3
**Contribution:** 3
**Rating:** 6
**Confidence:** 3

**Summary:**

This paper examines the impact of image compression on MLLMs, showing compression distortion harms cross-level features with low-level details with high-level semantics. Motivated by this insight, the authors propose CoTAM, a codec tailored for MLLMs that adaptively allocates bits based on shallow CLIP attention and reconstructs multi-level features via a lightweight adapter and multi-level loss. The encoder leverages semantic priors to prioritize important regions, while the decoder balances low-level fidelity and high-level semantics. CoTAM is compatible with both image and video MLLMs and supports high-resolution inputs through hierarchical guidance. Experiments show up to 36% bitrate savings without performance loss, outperforming both human-centric and machine-centric codecs.

**Strengths:**

I think the paper has following strengths:

1. Interesting finding and motivations: The paper provides analysis showing that compression artifacts disrupt cross-level features (merging low-level details with high-level semantics) rather than uniformly harming all visual cues. This finding is well-supported by attention-distance metrics and cosine-similarity experiments across multiple vision encoders.

2. Good feasibility: The method wraps existing neural codecs with only a lightweight adapter (1 transformer block) plus a negligible 128-bit metadata overhead, making it easy to integrate into production pipelines without retraining.

**Weaknesses:**

However, I have some small concerns about the paper:

1. Task-Specific Bias in Evaluation: Most of the benchmark tasks are on high-level semantic understanding tasks, what if the author benchmarks some low-level vision-related tasks besides OCR? I'm curious about that.

2. I'm doubtful about the scalability part. CoTAM’s CLIP-guided encoder relies on shallow-layer attention maps (8×8 grid) for bit allocation. While this works for 336×336 inputs, its scalability to higher resolutions (e.g., 4K) or longer video sequences is unclear. The hierarchical guidance extension (Section 3.4) is only briefly tested, with no ablation on how local/global attention fusion scales computationally. A 128-bit overhead for 336×336 images may balloon for larger inputs.

**Questions:**

In general, I'm satified with the paper, but small concern remains. Please see the weaknesses section for details.

---

> ### Author Response · Authors · 2025-11-19
> **Response to Reviewer iGt4**
>
> 1. **Clarify for Task-Specific Bias in Evaluation**
>
>     We thank the reviewer for this insightful question. We would like to clarify that our benchmark evaluation, in fact, covers a range of tasks that depend heavily on low-level visual information besides just OCR.
>     For example, tasks like image quality, OCR, position, and attribute recognition (color, shape) are all critically dependent on the precise fidelity of low-level details such as sharp edges, contours, and accurate color values. The results in Figure 1 confirm that the human-centric codec significantly outperforms the machine-centric method on these exact tasks. This reinforces our conclusion that the human-centric codec excels at preserving low-level information. The "Structuralized Image-text Understanding" tasks in Figures 22 and 23, which similarly rely on high-fidelity details for parsing layouts, further corroborate this point.
>
> 2. **Clarify for High-resolution Strategy**
>
> * "Clarify High-Resolution Strategy": Our high-resolution strategy processes the image through two complementary pathways before generating the final guidance map. In the local path, the image is cropped into sub-patches (e.g., 336 $\times$ 336), where each patch is processed to extract an 8 $\times$ 8 map. These maps are then concatenated to preserve spatial details. Simultaneously, if global attention is employed, the full image is downsampled to the patch size to extract a global map, which is subsequently resized to align with the concatenated local map. The results from both paths are summed, followed by three-level quantization.
>
>     For instance, a 672 $\times$ 672 input yields a 16 $\times$ 16 concatenated map from the local path. The global map is resized to 16 $\times$ 16 and added to this base, resulting in a final 16 $\times$ 16 guidance map.
>
> * "Why 8 $\times$ 8 grid":
> As shown in the table below, we also test the high-resolution MME performance using different grid sizes. Using an 8 $\times$ 8 grid can achieve better BD-Rate than a 24 $\times$ 24 grid. This is because higher resolution grids have greater bpp overhead (more than 0.01 bpp).
> Due to this considerations, we used an 8 $\times$ 8 grid for each sub-patch in the paper.
>
>     | sub-Patch Grid Size | BD-Rate $\downarrow$ | All Grids Bpp Overhead |
>     |-----------------|---------|---------|
>     | 4 $\times$ 4           | -61.12% | 0.0003    |
>     | 8 $\times$ 8           | -70.16% | 0.0010    |
>     | 12 $\times$ 12         | -69.94% | 0.0026    |
>     | 24 $\times$ 24         | -69.37% | 0.0102    |
>
> * "How local/global attention fusion scales computationally": Because the attention of the patches obtained from local and global scores can be computed in parallel, and we only use half-precision to compute the attention of the first three layers of CLIP, the computation of attention does not bring much overhead compared to the codec itself. Here we provide the encoding and decoding time of codecs at a 1344 $\times$ 1344 or 1008 $\times$ 1008 image size. It can be seen that local+global only has 2% more encoding time than global.
>
>     1344x1344 size:
>
>     | Method         | Encode | Decode |
>     |----------------|--------|--------|
>     | Local          | 0.558  | 0.294  |
>     | Global         | 0.549  | 0.294  |
>     | Local + Global | 0.559  | 0.294  |
>
>     1008x1008 size:
>
>     |   Method             | Encode | Decode |
>     |----------------|--------|--------|
>     | Local          | 0.381  |  0.200 |
>     | Global         | 0.374  |  0.200 |
>     | Local + Global | 0.382  |  0.200 |
>
> * "High-resolution guidance maps bit overhead": At high resolution, because multiple sub-patches need a grid, the total bit overhead increases, but as shown in the table below, the overall bpp (bits per pixel) remains constant at approximately 0.001 bpp which is alway negligible at any resolution.
>
>     | Input Resolution | Total Grid Size | Guidance Bit Overheads | All Grids Bpp Overheads |
>     |------------------|-----------------|------------------------|---------------|
>     | 336x336          | 8 $\times$ 8           | 128 bits               | 0.0010        |
>     | 672x672          | 16 $\times$ 16         | 512 bits               | 0.0010        |
>     | 1344x1344        | 32 $\times$ 32         | 2048 bits              | 0.0010        |

---

> > ### Comment · Reviewer_iGt4 · 2025-11-27
> > **Response to the authors**
> >
> > Thanks the author for the detailed explanations. I have also read other reviewers' comments, and most of the responses have addressed my concerns. I will keep my current positive score.

---

### Author Response · Authors · 2025-12-01
**Rebuttal Summary**

# Rebuttal Status & Timeline

We would like to thank all reviewers for their time and constructive comments. We provide a summary of the discussion phase here.

During the discussion phase, we engaged in active dialogue with all four reviewers. **Importantly, all reviewer discussions were concluded, and final consensus was reached, before the recent OpenReview Data Leak issue.**

Our rebuttal successfully addressed the reviewers' concerns, resulting in a score improvement from **6/6/8/4** to **6/6/8/6**.

* **Reviewer `g6n1`**: Raised their score from **4** to **6** after we provided additional comparisons with traditional codecs and validated robustness across diverse MLLMs.
* **Reviewers `iGt4`, `yANt`, and `XBDN`**: Maintained their positive scores (**6**, **6**, and **8**), explicitly stating that their concerns were resolved.

---

# Rebuttal Summary

We addressed the key questions raised across the reviews as follows:

### 1. Demonstrated robustness across diverse backbones and codecs
*(Reviewer `g6n1`, `yANt`)*

We addressed concerns about backbone diversity by providing additional experiments on distinct LLM architectures (Vicuna, Qwen2, InternLM2) and Vision Encoders (SigLIP, InternViT).
* Results confirm CoTAM is codec-agnostic and robust.
* Additionally, we added comparisons with traditional codecs (VTM-18.1), demonstrating that CoTAM achieves significant bitrate savings.

### 2. Validated scalability and justified implementation choices
*(Reviewer `iGt4`, `yANt`)*

* We clarified our high-resolution strategy (Local + Global pathways) to demonstrate scalability.
* We also justified the choice of the 8x8 guidance grid, proving via ablation studies that it offers the optimal trade-off between BPP overhead and performance compared to finer grids.

### 3. Substantiated the novelty and justification of our Three-Stage Theory
*(Reviewer `yANt`, `XBDN`)*

We clarified that our "Three-Stage Theory" extends beyond known ViT properties by pinpointing "cross-level feature synthesis" as the specific vulnerability during compression. This insight is supported by our quantitative analysis (attention distance, PCA).

### 4. Ensured fair comparison and expanded evaluation metrics
*(Reviewer `XBDN`, `iGt4`)*

* We validated CoTAM’s performance on low-level tasks (e.g., OCR, attribute recognition) to address concerns about task-specific bias.
* Furthermore, we clarified our experimental setup regarding fixed quality parameters versus fixed bitrates, satisfying Reviewer `XBDN`'s inquiry regarding fair comparison.

---

We are grateful that the reviewers found our responses satisfactory and that the discussions concluded on a positive note. We respectfully submit this summary to assist you in your final assessment.

---

### Meta-Review · Area_Chair_cyw1 · 2026-01-02

**Summary:**

All reviewers agree to accept the paper. The score raises from 6/6/8/4 to 6/6/8/6 during discussion.

The topic combines neural codecs with MLLM, which is very interesting and has practical values. The core methodology is well-motivated. The idea of first analyzing the internal failure point of MLLMs under compression (the vulnerability of cross-level features) and then designing a codec (multi-level loss, semantic guidance) to specifically protect that failure point is logical and elegant.

The experimental validation is a major strength. The authors evaluate on a wide and diverse set of MLLM benchmarks, which is essential for general-purpose MLLM compression. The reported BD-rate savings of ~36% (Table 2) are substantial and clearly demonstrate the method's effectiveness over strong, relevant baselines.

**Reviewer Concerns:**

Concerns addressed:
Need to benchmark on both high-level semantic understanding tasks and low-level vision-related tasks.
The scalability to higher resolutions (e.g., 4K) or longer video sequences.
The "three-stage information flow" (Sec 2.2.1) is more like a re-characterization of known ViT properties in the context of compression, rather than a fundamental new insight into MLLMs.
The 'Shallow CLIP Guidance' (Sec 3.2) is implementation-specific, creating a dependency that the paper doesn't fully resolve.
How the bitrate control is conducted, per dataset or per image.



Concerns not fully addressed:
The results of MME on VTM are not very good and the work targets relatively few downstream tasks.

**Reviewer Scores:**

iGt4 kept 6.
yANt kept 6.
xBDN might keep 8.
g6n1 might increase from 4 to 6 as he stated.

---

### Decision · Program_Chairs · 2026-01-26

Accept (Poster)